

COMPUTO

ISSN 2824-7795

# AdaptiveConformal: An R Package for Adaptive Conformal Inference

Herbert Susmann ⬤[1]    CEREMADE (UMR 7534), Université Paris-Dauphine PSL, Place du Maréchal de Lattre de Tassigny, Paris, 75016, France

Antoine Chambaz ⬤    Université Paris Cité, CNRS, MAP5, F-75006 Paris, France

Julie Josse ⬤    Inria PreMeDICaL team, Idesp, Université de Montpellier

Date published: 2024-06-13    Last modified: 2024-06-13

### Abstract

Conformal Inference (CI) is a popular approach for generating finite sample prediction intervals based on the output of any point prediction method when data are exchangeable. Adaptive Conformal Inference (ACI) algorithms extend CI to the case of sequentially observed data, such as time series, and exhibit strong theoretical guarantees without having to assume exchangeability of the observed data. The common thread that unites algorithms in the ACI family is that they adaptively adjust the width of the generated prediction intervals in response to the observed data. We provide a detailed description of five ACI algorithms and their theoretical guarantees, and test their performance in simulation studies. We then present a case study of producing prediction intervals for influenza incidence in the United States based on black-box point forecasts. Implementations of all the algorithms are released as an open-source R package, AdaptiveConformal, which also includes tools for visualizing and summarizing conformal prediction intervals.

*Keywords:* Conformal inference, Adaptive conformal inference, time series, R

# Contents

---

[1]Corresponding author: herbps10@gmail.com

# 1 Introduction

Conformal Inference (CI) is a family of methods for generating finite sample prediction intervals around point predictions when data are exchangeable (Vovk, Gammerman, and Shafer 2005; Shafer and Vovk 2008; Angelopoulos and Bates 2023). The input point predictions can be derived from any prediction method, making CI a powerful tool for augmenting black-box prediction algorithms with prediction intervals. Classical CI methods are able to yield marginally valid intervals with only the assumption that the joint distribution of the data does not change based on the order of the observations (that is, they are exchangeable). However, in many real-world settings data are not exchangeable: for example, time series data usually cannot be assumed to be exchangeable due to temporal dependence. A recent line of research examines the problem of generating prediction intervals for observations that are observed online (that is, one at a time) and for which exchangeability is not assumed to hold (Gibbs and Candes 2021; Zaffran et al. 2022; Gibbs and Candès 2022; Bhatnagar et al. 2023). The methods from this literature, which we refer to generally as *Adaptive Conformal Inference* (ACI) algorithms, work by adaptively adjusting the width of the generated prediction intervals in response to the observed data.

Informally, suppose a sequence of outcomes $y_t \in \mathbb{R}$, $t = 1, \ldots, T$ are observed one at a time. Before

seeing each observation, we have at our disposal a point prediction $\hat{\mu}_t \in \mathbb{R}$ that can be generated by any method. Our goal is to find an algorithm for producing prediction intervals $[\ell_t, u_t]$, $\ell_t \leq u_t$ such that, in the long run, the observations $y_t$ fall within the corresponding prediction intervals roughly $\alpha \times 100\%$ of the time: that is, $\lim_{T \to \infty} 1/T \sum_{t=1}^{T} \mathbb{I}\{y_t \in [\ell_t, u_t]\} = \alpha$. The original ACI algorithm (Gibbs and Candes 2021) is based on a simple idea: if the previous prediction interval at time $(t-1)$ did not cover the true observation, then the next prediction interval at time $t$ is made slightly wider. Conversely, if the previous prediction interval did include the observation, then the next prediction interval is made slightly narrower. It can be shown that this procedure yields prediction intervals that in the long run cover the true observations the desired proportion of the time.

The main tuning parameter of the original ACI algorithm is a learning rate that controls how fast prediction interval width changes. If the learning rate is too low, then the prediction intervals will not be able to adapt fast enough to shifts in the data generating distribution; if it is too large, then the intervals will oscillate widely. The critical dependence of the original ACI algorithm on proper choice of its learning rate spurred subsequent research into meta-algorithms that learn the correct learning rate (or an analogue thereof) in various ways, typically drawing on approaches from the online learning literature. In this paper, we present four such algorithms: Aggregated ACI (AgACI, Zaffran et al. 2022), Dynamically-tuned Adaptive ACI (DtACI, Gibbs and Candès 2022), Scale-Free Online Gradient Descent (SF-OGD, Bhatnagar et al. 2023), and Strongly Adaptive Online Conformal Prediction (SAOCP, Bhatnagar et al. 2023). We note that the adaption of conformal inference techniques is an active area of research and the algorithms we focus on in this work are not exhaustive; see among others Feldman et al. (2023), Bastani et al. (2022), Xu and Xie (2021), Xu and Xie (2023), Angelopoulos, Barber, and Bates (2024), Zhang, Bombara, and Yang (2024), and Gasparin and Ramdas (2024).

Our primary practical contribution is an implementation of each algorithm in an open source `R` package, `AdaptiveConformal`, which is available at https://github.com/herbps10/AdaptiveConformal. The package also includes routines for visualization and summary of the prediction intervals. We note that Python versions of several algorithms were also made available by Zaffran et al. (2022) and Bhatnagar et al. (2023), but to our knowledge this is the first package implementing them in `R`. In addition, several `R` packages exist for conformal inference in other contexts, including `conformalInference` focusing on regression (Tibshirani et al. 2019), `conformalInference.fd`, with methods for functional responses (Diquigiovanni et al. 2022), and `cfcausal` for causal inference related functionals (Lei and Candès 2021). Our second practical contribution is to compare the performance of the algorithms in simulation studies and in a case study generating prediction intervals for influenza incidence in the United States based on black-box point forecasts.

The rest of the paper unfolds as follows. In Section 2, we present a unified theoretical framework for analyzing the ACI algorithms based on the online learning paradigm. In Section 3 we provide descriptions of each algorithm along with their known theoretical properties. In Section 5 we compare the performance of the algorithms in several simulation studies. Section 6 gives a case study based on forecasting influenza in the United States. Finally, Section 7 provides a discussion and ideas for future research in this rapidly expanding field.

## 2 Theoretical Framework

*Notation*: for any integer $N \geq 1$ let $[\![N]\!] := \{1, \ldots, N\}$. Let $\mathbb{I}$ be the indicator function. Let $\nabla f$ denote the gradient (subgradient) of the differentiable (convex) function $f$.

We consider an online learning scenario in which we gain access to a sequence of observations $(y_t)_{t \geq 1}$ one at a time (see Cesa-Bianchi and Lugosi (2006) for an comprehensive account of online learning theory). Fix $\alpha \in (0, 1)$ to be the target empirical coverage of the prediction intervals. The goal is

to output at time $t$ a prediction interval for the unseen observation $y_t$, with the prediction interval generated by an *interval construction function* $\hat{C}_t$. Formally, let $\hat{C}_t$ be a function that takes as input a parameter $\theta_t \in \mathbb{R}$ and outputs a closed prediction interval $[\ell_t, u_t]$. The interval construction function must be nested: if $\theta' > \theta$, then $\hat{C}_t(\theta) \subseteq \hat{C}_t(\theta')$. In words, larger values of $\theta$ imply wider prediction intervals. The interval constructor is indexed by $t$ to emphasize that it may use other information at each time point, such as a point prediction $\hat{\mu}_t \in \mathbb{R}$. We make no restrictions on how this external information is generated.

Define $r_t := \inf\{\theta \in \mathbb{R} : y_t \in \hat{C}_t(\theta)\}$ to be the *radius* at time $t$. The radius is the smallest possible $\theta$ such that the prediction interval covers the observation $y_t$. A key assumption for the theoretical analysis of several of the algorithms is that the radii are bounded:

**Assumption**: there exists a finite $D > 0$ such that $r_t < D$ for all $t$.

If the outcome space is bounded, then $D$ can be easily chosen to cover the entire space. Next, we describe two existing definitions of interval construction functions.

## 2.1 Linear Intervals

A simple method for forming the prediction intervals is to use the parameter $\theta_t$ to directly define the width of the interval. Suppose that at each time $t$ we have access to a point prediction $\hat{\mu}_t \in \mathbb{R}$. Then we can form a symmetric prediction interval around the point estimate using

$$\theta \mapsto \hat{C}_t(\theta) := [\hat{\mu}_t - \theta, \hat{\mu}_t + \theta].$$

We refer to this as the *linear interval constructor*. Note that in this case, the radius is simply the absolute residual $r_t = |\hat{\mu}_t - y_t|$.

## 2.2 Quantile Intervals

The original ACI paper proposed constructing intervals based on the previously observed residuals (Gibbs and Candes 2021). Let $S : \mathbb{R}^2 \to \mathbb{R}$ be a function called a *nonconformity score*. A popular choice of nonconformity score is the absolute residual: $(\mu, y) \mapsto S(\mu, y) := |\mu - y|$. Let $s_t := S(\hat{\mu}_t, y_t)$ be the nonconformity score of the $t$th-observation. The quantile interval construction function is then given by

$$\hat{C}_t(\theta_t) := [\hat{\mu}_t - \text{Quantile}(\theta, \{s_1, \dots, s_{t-1}\}), \hat{\mu}_t + \text{Quantile}(\theta, \{s_1, \dots, s_{t-1}\})]$$

where $\text{Quantile}(\theta, A)$ denotes the empirical $\theta$-quantile of the elements in the set $A$. Note that $\hat{C}_t$ is indeed nested in $\theta_t$ because the Quantile function is non-decreasing in $\theta$. Note we define $\hat{C}_t(1) = \max\{s_1, \dots, s_{t-1}\}$ rather than $\hat{C}_t(1) = \infty$ in order to avoid practical problems with trivial prediction intervals (Zaffran et al. 2022). Note that we can always choose $D = 1$ to satisfy the outcome boundedness assumption given above.

We focus on the above definition for the quantile interval construction function which is designed to be symmetric around the point prediction $\hat{\mu}_t$. However, we note it is possible to take a more general definition, such as

$$\hat{C}_t(\theta_t) := \{y : S(\hat{\mu}_t, y) \le \text{Quantile}(, \{s_1, \dots, s_{t-1}\})\}$$

Such an approach allows for prediction intervals that may not be centered on $\hat{\mu}_t$.

Our proposed `AdaptiveConformal` package takes the absolute residual as the default nonconformity score, although the user may also specify any custom nonconformity score by supplying it as an `R` function.

## 2.3 Online Learning Framework

We now introduce a loss function that defines the quality of a prediction interval with respect to a realized observation. Define the *pinball loss $L^\alpha$* as

$$(\theta, r) \mapsto L^\alpha(\theta, r) := \begin{cases} (1 - \alpha)(\theta - r), & \theta \geq r \\ \alpha(r - \theta), & \theta < r. \end{cases}$$

The way in which the algorithm gains access to the data and incurs losses is as follows:

- Sequentially, for $t = 1, \ldots, T$:
    - Predict radius $\theta_t$ and form prediction interval $\hat{C}_t(\theta_t)$.
    - Observe true outcome $y_t$ and calculate radius $r_t$.
    - Record $\text{err}_t := \mathbb{I}[y_t \notin \hat{C}_t(\theta_t)]$.
    - Incur loss $L^\alpha(\theta_t, r_t)$.

This iterative procedure is at the core of the online learning theoretical framework in which theoretical results have been derived.

## 2.4 Assessing ACI algorithms

There are two different perspectives we can take in measuring the quality of an ACI algorithm that generates a sequence $(\theta_t)_{t \in [\![T]\!]}$. First, we could look at how close the empirical coverage of the generated prediction intervals is to the desired coverage level $\alpha$. Formally, define the empirical coverage as the proportion of observations that fell within the corresponding prediction interval: $\text{EmpCov}(T) := \frac{1}{T} \sum_{t=1}^{T}(1 - \text{err}_t)$. The coverage error is then given by

$$\text{CovErr}(T) := \text{EmpCov}(T) - \alpha.$$

The second perspective is to look at how well the algorithm controls the incurred pinball losses. Following the classical framework from the online learning literature, we define the *regret* as the difference between the cumulative loss yielded by a sequence $(\theta_t)_{t \in [\![T]\!]}$ versus the cumulative loss of the best possible fixed choice:

$$\text{Reg}(T) := \sum_{t=1}^{T} L^\alpha(\theta_t, r_t) - \inf_{\theta^* \in \mathbb{R}} \sum_{t=1}^{T} L^\alpha(\theta^*, r_t).$$

In settings of distribution shift, it may not be appropriate to compare the cumulative loss of an algorithm to a fixed competitor. As such, stronger notions of regret have been defined. The *strongly adaptive regret* is the largest regret over any subperiod of length $m \in [\![T]\!]$:

$$\text{SAReg}(T, m) := \max_{[\tau, \tau+m-1] \subseteq [\![T]\!]} \left( \sum_{t=\tau}^{\tau+m-1} L^\alpha(\theta_t, r_t) - \inf_{\theta^* \in \mathbb{R}} \sum_{t=\tau}^{\tau+m-1} L^\alpha(\theta^*, r_t) \right).$$

Both ways of evaluating ACI methods are important because targeting only one or the other can lead to algorithms that yield prediction intervals that are not practically useful. As a simple pathological example of only targeting the coverage error, suppose we wish to generate $\alpha = 50\%$ prediction intervals. We could choose to alternate $\theta$ between 0 and $\infty$, such that $\text{err}_t$ alternates between 0 and 1. The empirical coverage would then trivially converge to the desired level of 50%. However, the same algorithm would yield infinite regret (see Bhatnagar et al. (2023) for a more in-depth example of an scenario in which coverage is optimal but the regret grows linearly). On the other hand, an algorithm that has arbitrarily small regret may not yield good empirical coverage. Suppose the observations and point predictions are constant: $y_t = 1$ and $\hat{\mu}_t = 0$ for all $t \geq 1$. Consider a simple class of algorithms

that outputs constantly $\theta_t = \theta'$ for some $\theta' < 1$. With the linear interval construction function, the prediction intervals are then $\hat{C}_t(\theta_t) = [-\theta', \theta']$. The regret is given by $\text{Reg}(T) = 2T\alpha(1 - \theta')$, which approaches zero as $\theta'$ approaches 1. The empirical coverage is, however, always zero. In other words, the regret can be arbitrarily close to zero while at the same time the empirical coverage does not approach the desired level.

These simple examples illustrate that, unfortunately, bounds on the coverage error and bounds on the regret are not in general interchangeable. It is possible, however, to show equivalencies by either (1) making distributional assumptions on the data or (2) using additional information about how the algorithm produces the sequence $(\theta_t)_{t \in [\![T]\!]}$ (Bhatnagar et al. 2023).

It may also be informative to summarize a set of prediction intervals in ways beyond their coverage error or their regret. A common metric for prediction intervals is the *mean interval width*:

$$\text{MeanWidth}(T) := \frac{1}{T} \sum_{t=1}^{T} w_t,$$

where $w_t := u_t - \ell_t$ is the interval width at time $t$.

Finally, we introduce a metric that is intended to capture pathological behavior that can arise with ACI algorithms where the prediction intervals oscillate between being extremely narrow and extremely wide. Define the *path length* of prediction intervals generated by an ACI algorithm as

$$\text{PathLength}(T) := \sum_{t=2}^{T} |w_t - w_{t-1}|.$$

A high path length indicates that the prediction intervals were variable over time, and a low path length indicates the prediction intervals were stable.

# 3   Algorithms

Table 1: Summary of ACI algorithms. Only the theoretical guarantees discussed in this paper are shown for each algorithm.

| Algorithm |
| --- |
| **Adaptive Conformal Inference (ACI)** |
| - Tuning parameters: learning rate $\gamma$ |
| - Original interval constructor: quantile |
| - Theoretical guarantees: coverage error, regret |
| - Citation: Gibbs and Candes (2021) |
| **Aggregated Adaptive Conformal Inference (AgACI)** |
| - Tuning parameters: candidate learning rates $(\gamma_k)_{1 \leq k \leq K}$ |
| - Original interval constructor: quantile |
| - Citation: Zaffran et al. (2022) |
| **Dynamically-tuned Adaptive Conformal Inference (DtACI)** |
| - Tuning parameters: candidate learning rates $(\gamma_k)_{1 \leq k \leq K}$ |
| - Original interval constructor: quantile |
| - Theoretical guarantees: coverage error, strongly adaptive regret, dynamic regret |
| - Citation |
| **Scale-Free Online Gradient Descent (SF-OGD)** |
| - Tuning parameters: learning rate $\gamma$ or maximum radius $D$ |

As a simple running example to illustrate each algorithm, we simulate independently $T = 500$ values $y_1, \dots, y_T$ following

$$y_t \sim N(0, \sigma_t^2), \quad t \in [\![T]\!],$$

$$\sigma_t = \begin{cases} 0.2, & t \le 250, \\ 0.1, & t > 250. \end{cases}$$

For demonstration purposes we assume we have access to unbiased predictions $\hat{\mu}_t = 0$ for all $t \in [\![T]\!]$. Throughout we set the target empirical coverage to $\alpha = 0.8$.

## 3.1 Adaptive Conformal Inference (ACI)

---
**Algorithm 1** Adaptive Conformal Inference

---
1: **Input:** starting value $\theta_1$, learning rate $\gamma > 0$.
2: **for** $t = 1, 2, \dots, T$ **do**
3:     **Output:** prediction interval $\hat{C}_t(\theta_t)$.
4:     Observe $y_t$.
5:     Evaluate $\text{err}_t = \mathbb{I}[y_t \notin \hat{C}_t(\theta_t)]$.
6:     Update $\theta_{t+1} = \theta_t + \gamma(\text{err}_t - (1 - \alpha))$.
7: **end for**

---

The original ACI algorithm (Gibbs and Candes (2021); Algorithm 1 ) adaptively adjusts the width of the prediction intervals in response to the observations. The updating rule for the estimated radius can be derived as an online subgradient descent scheme. The subgradient of the pinball loss function with respect to $\theta$ is given by

$$\nabla L^{\alpha}(\theta, r) = \begin{cases} \{-\alpha\}, & \theta < r, \\ \{1 - \alpha\}, & \theta > r, \\ [-\alpha, 1 - \alpha], & \theta = r \end{cases}$$

It follows that, for all $\theta_t \in \mathbb{R}$ and $r_t \in \mathbb{R}$,

$$1 - \alpha - \text{err}_t \in \nabla L^{\alpha}(\theta_t, r_t).$$

This leads to the following update rule for $\theta$ based on subgradient descent:

$$\theta_{t+1} = \theta_t + \gamma(\text{err}_t - (1 - \alpha)),$$

where $\gamma > 0$ is a user-specified learning rate. For intuition, note that if $y_t$ fell outside of the prediction interval at time $t$ ($\text{err}_t = 1$) then the next interval is widened ($\theta_{t+1} = \theta_t + \gamma\alpha$). On the contrary, if $y_t$ fell within the interval ($\text{err}_t = 0$) then the next interval is shortened ($\theta_{t+1} = \theta_t - \gamma(1 - \alpha)$). The learning rate $\gamma$ controls how fast the width of the prediction intervals changes in response to the data.

### 3.1.1 Theoretical Guarantees

With the choice of the quantile interval structure, the ACI algorithm has the following finite sample bound on the coverage error (Gibbs and Candes (2021); Proposition 4.1). For all $\gamma > 0$ (and so long as $\gamma$ does not depend on $T$),

$$|\text{CovErr}(T)| \leq \frac{\max\{\theta_1, 1 - \theta_1\} + \gamma}{\gamma T}.$$

This result was originally shown for ACI with the choice of the quantile interval constructor, although it can also be extended to other interval constructors Feldman et al. (2023). More generally, the algorithm has the following coverage error bound in terms of the radius bound $D$ (Bhatnagar et al. 2023):

$$|\text{CovErr}(T)| \leq \frac{D + \gamma}{\gamma T}.$$

In addition, standard results for online subgradient descent yield the following regret bound with the use of the linear interval constructor, assuming that the true radii are bounded by $D$:

$$\text{Reg}(T) \leq \mathcal{O}(D^2/\gamma + \gamma T) \leq \mathcal{O}(D\sqrt{T}),$$

where the second inequality follows if the optimal choice (with respect to long-term regret) of $\gamma = D/\sqrt{T}$ is used (Bhatnagar et al. 2023). Taken together, these theoretical results imply that while the coverage error is guaranteed to converge to zero for any choice of $\gamma$, achieving sublinear regret requires choosing $\gamma$ more carefully. This highlights the importance of both ways of assessing ACI algorithms: if we only focused on controlling the coverage error, we might not achieve optimal control of regret, leading to intervals that are not practically useful.

### 3.1.2 Tuning Parameters

Therefore, the main tuning parameter is the learning rate $\gamma$. The theoretical bounds on the coverage error suggest setting a large $\gamma$ such that the coverage error decays quickly in $T$; however, in practice and setting $\gamma$ too large will lead to intervals with large oscillations as seen in Figure 1. This is quantified in the path length, which increases significantly as $\gamma$ increases, even though the empirical coverage remains near the desired value of 80%. On the other hand, setting $\gamma$ too small will lead to intervals that do not adapt fast enough to distribution shifts. Thus, choosing a good value for $\gamma$ is essential. However, the optimal choice $\gamma = D/\sqrt{T}$ cannot be used directly in practice unless the time series length $T$ is fixed in advance, or the so called "doubling trick" is used to relax the need to know $T$ in advance (Cesa-Bianchi and Lugosi (2006); Section 2.3).

The theoretical results guaranteeing the performance of the ACI algorithm do not depend on the choice of starting value $\theta_1$, and thus in practice any value can be chosen. Indeed, the effect of the choice of $\theta_1$ decays over time as a function of the chosen learning rate. In practice, substantive prior information can be used to pick a reasonable starting value. By default, the `AdaptiveConformal` package sets $\theta_1 = \alpha$ when the quantile interval predictor is used, and $\theta_1 = 0$ otherwise, although in both cases the user can supply their own starting value. The behavior of the early prediction intervals in the examples (Figure 1) is driven by the small number of residuals available, which makes the output of the quantile interval constructor sensitive to small changes in $\theta$. In practice, a warm-up period can be used before starting to produce prediction intervals so that the quantiles of the residuals are more stable.

## 3.2 Aggregated Adaptive Conformal Inference (AgACI)

The Aggregated ACI (AgACI; Algorithm 2 ) algorithm solves the problem of choosing a learning rate for ACI by running multiple copies of the algorithm with different learning rates, and then separately

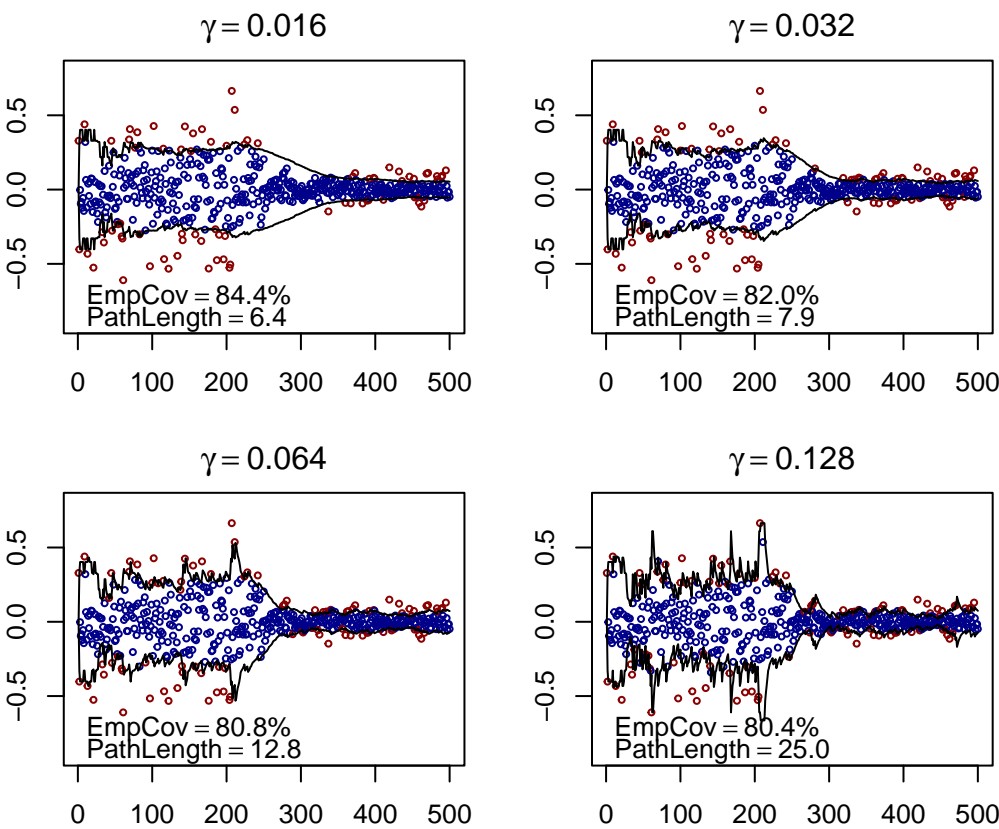

Figure 1: Example 80% prediction intervals from the ACI algorithm for different choices of learning rate $\gamma$ and with $\theta_1 = 0.8$. Blue and red points are observations that fell inside and outside the prediction intervals, respectively.

---
**Algorithm 2** Aggregated Adaptive Conformal Inference
---
1: **Input:** candidate learning rates $(\gamma_k)_{1 \le k \le K}$, starting value $\theta_1$.
2: Initialize lower and upper BOA algorithms $\mathscr{B}^\ell := \texttt{BOA}(\alpha \leftarrow (1-\alpha)/2)$ and $\mathscr{B}^u := \texttt{BOA}(\alpha \leftarrow (1-(1-\alpha)/2))$.
3: **for** $k = 1, \dots, K$ **do**
4:     Initialize ACI $\mathscr{A}_k = \texttt{ACI}(\alpha \leftarrow \alpha, \gamma \leftarrow \gamma_k, \theta_1 \leftarrow \theta_1)$.
5: **end for**
6: **for** $t = 1, 2, \dots, T$ **do**
7:     **for** $k = 1, \dots, K$ **do**
8:         Retrieve candidate prediction interval $[\ell_t^k, u_t^k]$ from $\mathscr{A}_k$.
9:     **end for**
10:     Compute aggregated lower bound $\tilde{\ell}_t := \mathscr{B}^\ell((\ell_t^k : k \in \{1, \dots, K\}))$.
11:     Compute aggregated upper bound $\tilde{u}_t := \mathscr{B}^u((u_t^k : k \in \{1, \dots, K\}))$.
12:     **Output:** prediction interval $[\tilde{\ell}_t, \tilde{u}_t]$.
13:     Observe $y_t$.
14:     **for** $k = 1, \dots, K$ **do**
15:         Update $\mathscr{A}_k$ with observation $y_t$.
16:     **end for**
17:     Update $\mathscr{B}^\ell$ with observed outcome $y_t$.
18:     Update $\mathscr{B}^u$ with observed outcome $y_t$.
19: **end for**
---

combining the lower and upper interval bounds using an online aggregation of experts algorithm (Zaffran et al. 2022). That is, one aggregation algorithm seeks to estimate the lower $(1-\alpha)/2$ quantile, and the other seeks to estimate the upper $1 - (1-\alpha)/2$ quantile. Zaffran et al. (2022) experimented with multiple online aggregation algorithms, and found that they yielded similar results. Thus, we follow their lead in using the Bernstein Online Aggregation (BOA) algorithm as implemented in the `opera` R package (Wintenberger 2017; Gaillard et al. 2023). BOA is an online algorithm that forms predictions for the lower (or upper) prediction interval bound as a weighted mean of the candidate ACI prediction interval lower (upper) bound, where the weights are determined by each candidate's past performance with respect to the quantile loss. As a consequence, the prediction intervals generated by AgACI are not necessarily symmetric around the point prediction, as the weights for the lower and upper bounds are separate.

### 3.2.1 Theoretical Gaurantees

AgACI departs from our main theoretical framework in that it does not yield a sequence $(\theta_t)_{t \in [\![T]\!]}$ whose elements yield prediction intervals via a set construction function $\hat{C}_t$. Rather, the upper and lower interval bounds from a set of candidate ACI algorithms are aggregated separately. Thus, theoretical results such as regret bounds similar to those for the other algorithms are not available. It would be possible, however, to establish regret bounds for the pinball loss applied separately to the lower and upper bounds of the prediction intervals. It is unclear, however, how to convert such regret bounds into a coverage bound.

### 3.2.2 Tuning Parameters

The main tuning parameter for AgACI is the set of candidate learning rates. Beyond necessitating additional computational time, there is no drawback to having a large grid. As a default, `AdaptiveConformal` uses learning rates $\gamma \in \{0.001, 0.002, 0.004, 0.008, 0.016, 0.032, 0.064, 0.128\}$. As a basic check, we can look at the weights assigned to each of the learning rates. If large weights are

given to the smallest (largest) learning rates, it is a sign that smaller (or larger) learning rates may perform well. In addition each of the candidate ACI algorithms requires a starting value, which can be set to any value as discussed in the ACI section. Figure 2 illustrates AgACI applied to the running example with two sets of learning grids. The first grid is $\gamma = \{0.032, 0.064, 0.128, 0.256\}$, and the second grid includes the additional values $\{0.008, 0.016\}$. For the first grid, we can see that for the lower bound AgACI assigns high weight to the lowest learning rate ($\gamma = 0.032$). The second grid yields weights that are less concentrated on a single learning rate, and the output prediction intervals are smoother.

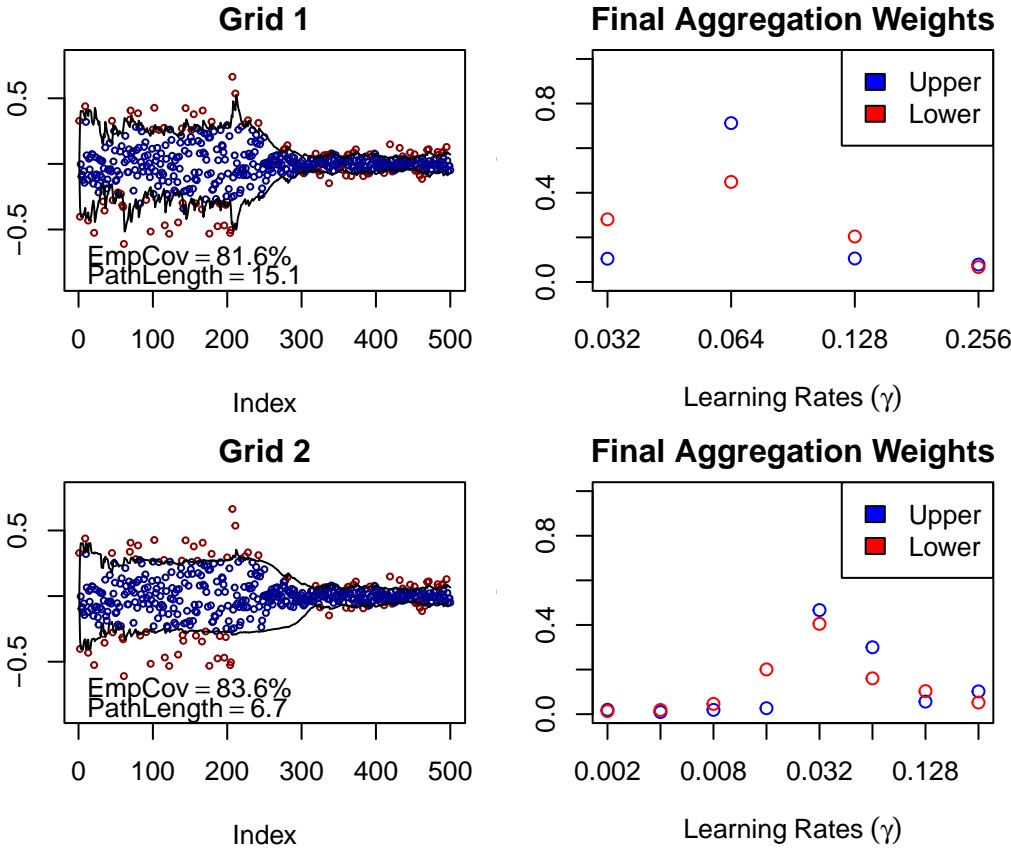

Figure 2: Example 80% prediction intervals from the AgACI algorithm with starting values $\theta_1 = 0.8$ and two different learning rate grids. In the left column, blue and red points are observations that fell inside and outside the prediction intervals, respectively.

## 3.3 Dynamically-tuned Adaptive Conformal Inference (DtACI)

The Dynamically-tuned Adaptive Conformal Inference (DtACI; Algorithm 3 ) algorithm was developed by the authors of the original ACI algorithm in part to address the issue of how to choose the learning rate parameter $\gamma$. In this respect the goal of the algorithm is similar to that of AgACI, although it is achieved slightly differently. DtACI also aggregates predictions from multiple copies of ACI run with different learning rates, but differs in that it directly aggregates the estimated radii emitted from each algorithm based on their pinball loss (Gibbs and Candès 2022) using an exponential reweighting scheme (Gradu, Hazan, and Minasyan 2023). As opposed to AgACI, this construction allows for more straightforward development of theoretical guarantees on the algorithm's performance, because the upper and lower bounds of the intervals are not aggregated separately.

---

**Algorithm 3** Dynamically-tuned Adaptive Conformal Inference

---

1: **Input:** starting value $\theta_1$, candidate learning rates $(\gamma_k)_{1 \leq k \leq K}$, parameters $\sigma, \eta$.
2: **for** $k = 1, \ldots, K$ **do**
3:      Initialize expert $\mathcal{A}_k = \texttt{ACI}(\alpha \leftarrow \alpha, \gamma \leftarrow \gamma_k, \theta_1 \leftarrow \theta_1)$.
4: **end for**
5: **for** $t = 1, 2, \ldots, T$ **do**
6:      Define $p_t^k := p_t^k / \sum_{i=1}^{K} p_t^i$, for all $1 \leq k \leq K$.
7:      Set $\theta_t = \sum_{k=1}^{K} \theta_t^k p_t^k$.
8:      **Output:** prediction interval $\hat{C}_t(\theta_t)$.
9:      Observe $y_t$ and compute $r_t$.
10:      $\bar{w}_t^k \leftarrow p_t^k \exp(-\eta L^\alpha(\theta_t^k, r_t))$, for all $1 \leq k \leq K$.
11:      $\bar{W}_t \leftarrow \sum_{i=1}^{K} \bar{w}_t^i$.
12:      $p_{t+1}^k \leftarrow (1 - \sigma)\bar{w}_t^k + \bar{W}_t \sigma / K$.
13:      Set $\text{err}_t := \mathbb{I}[y_t \notin \hat{C}_t(\theta_t)]$.
14:      **for** $k = 1, \ldots, K$ **do**
15:          Update ACI $\mathcal{A}_k$ with $y_t$ and obtain $\theta_{t+1}^k$.
16:      **end for**
17: **end for**

---

### 3.3.1 Theoretical Guarantees

DtACI was originally proposed with the choice of the quantile interval constructor. DtACI has the following strongly-adaptive regret bound (Bhatnagar et al. 2023): for all $\eta > 0$ and subperiod lengths $m$,

$$\text{SAReg}(T, m) \leq \widetilde{\mathcal{O}}(D^2/\eta + \eta m).$$

If $m$ is fixed a-priori, then choosing $\eta = D/\sqrt{m}$ yields a strongly adaptive regret bound of order $\widetilde{\mathcal{O}}(D\sqrt{m})$ (for a single choice of $m$). Practically, this result implies that, if we know in advance the time length for which we would like to control the regret, it is possible to choose an optimal tuning parameter value. However, we cannot control the regret simultaneously for all possible time lengths.

To establish a bound on the coverage error, the authors investigated a slightly modified version of DtACI in which $\theta_t$ is chosen randomly from the candidate $\theta_{t_k}$ with weights given by $p_{t,k}$, instead of taking a weighted average. This is a common trick used in the literature as it facilitates theoretical analysis. In practice, the authors comment that this randomized version of DtACI and the deterministic version lead to very similar results. The coverage error result also assumes that the hyperparameters can change over time: that is, we have $t$-specific $\eta_t$ and $\sigma_t$, rather than fixed $\eta$ and $\sigma$. The coverage error then has the following bound (Gibbs and Candès (2022); Theorem 3.2), where $\gamma_{\min}$ and $\gamma_{\max}$ are the smallest and largest learning rates in the grid, respectively:

$$|\text{CovErr}(T)| \leq \frac{1 + 2\gamma_{\max}}{T\gamma_{\min}} + \frac{(1 + 2\gamma_{\max})^2}{\gamma_{\min}} \frac{1}{T} \sum_{t=1}^{T} \eta_t \exp(\eta_t(1 + 2\gamma_{\max})) + 2\frac{1 + \gamma_{\max}}{\gamma_{\min}} \frac{1}{T} \sum_{t=1}^{T} \sigma_t.$$

Thus, if $\eta_t$ and $\sigma_t$ both converge to zero as $t \to \infty$, then the coverage error will also converge to zero. In addition, under mild distributional assumptions the authors provide a type of short-term coverage error bound for arbitrary time spans, for which we refer to (Gibbs and Candès 2022).

We note one additional result established by Gibbs and Candès (2022) (their Theorem 3.1) on a slightly different dynamic regret bound in terms of the pinball loss, as it informs the choice of tuning parameters. Let $\gamma_{\max} = \max_{1 \leq k \leq K} \gamma_k$ be the largest learning rate in the grid and assume that $\gamma_1 < \gamma_2 < \cdots < \gamma_K$ with $\gamma_{k+1}/\gamma \leq 2$ for all $1 \leq k < K$. Then, for any interval $I = [r, s] \subseteq [\![T]\!]$ and any

sequence $\theta_r^*, \ldots, \theta_s^*$, under the assumption that $\gamma_k \geq \sqrt{1 + 1/|I|}$,

$$\frac{1}{|I|} \sum_{t=r}^{s} \mathbb{E}[L^\alpha(\theta_t, r_t)] - \frac{1}{|I|} \sum_{t=r}^{s} L^\alpha(\theta_t, \theta_t^*) \leq \frac{\log(k/\sigma) + 2\sigma|I|}{\eta|I|} + \frac{\eta}{|I|} \sum_{t=r}^{s} \mathbb{E}[L^\alpha(\theta_t, r_t)^2]$$
$$+ 2\sqrt{3}(1 + \gamma_{\max})^2 \max \left\{ \sqrt{\frac{\sum_{t=r+1}^{s} |\theta_t^* - \theta_{t-1}^*| + 1}{|I|}}, \gamma_1 \right\},$$

where the expectation is over the randomness in the randomized version of the algorithm. Here the time interval $I$ (with length $|I|$) is comparable to the time period length $m$ for the strongly adaptive regret. The parameter $|I|$, the time interval of interest for which we would like to control, can be chosen arbitrarily. This dynamic regret bound can be converted to a strongly adaptive regret bound by choosing $\theta_t^*$ to be constant.

### 3.3.2 Tuning parameters

The recommended settings for the tuning parameters depend on choosing a time interval length $|I|$ for which we would like to control the pinball loss. The choice of $|I|$ can be chosen arbitrarily. For the tuning parameter $\sigma$, the authors suggest the optimal choice (with respect to the dynamic regret) $\sigma = 1/(2|I|)$. Choosing $\eta$ is more difficult. The authors suggest the following choice for $\eta$, which they show is optimal if there is in fact no distribution shift:

$$\eta = \sqrt{\frac{3}{|I|}} \sqrt{\frac{\log(K \cdot |I|) + 2}{(\alpha)^2(1 - \alpha)^3 + (1 - \alpha)^2 \alpha^3}}.$$

Note that this choice is optimal only for the quantile interval constructor, for which $\theta_t$ is a quantile of previous nonconformity scores. As an alternative, the authors point out that $\eta$ can be learned in an online fashion using the update rule

$$\eta_t := \sqrt{\frac{\log(|I|K) + 2}{\sum_{s=t-|I|}^{t-1} L^\alpha(\theta_s, r_s)}}.$$

Both ways of choosing $\eta$ led to very similar results in the original author's empirical studies. In our proposed `AdaptiveConformal` package, the first approach is used when the quantile interval construction function is chosen, and the latter approach for the linear interval construction function.

Figure 3 illustrates DtACI with the quantile interval construction function and with the learning rate grid $\gamma \in \{0.001, 0.002, 0.004, 0.008, 0.016, 0.032, 0.064, 0.128\}$. The tuning parameter $\eta$ was set to 0.001, 1, and 100 to show how the algorithm responds to extreme choices of the parameter, and to $\eta \approx 3.19$ according to the optimal choice recommendation with $I = 100$ as described in the previous section. The results show that, in this simple example, high values of $\eta$ may lead to intervals that are too reactive to the data, as seen in the longer path length. The algorithm appears more robust, however, to small choices of $\eta$.

## 3.4 Scale-Free Online Gradient Descent (SF-OGD)

Scale-Free Online Gradient Descent (SF-OGD; Algorithm 4 ) is a general algorithm for online learning proposed by Orabona and Pál (2018). The algorithm updates $\theta_t$ with a gradient descent step where the learning rate adapts to the scale of the previously observed gradients. SF-OGD was first used in the context of ACI as a sub-algorithm for SAOCP (described in the next section). However, it was found to have good performance by itself (Bhatnagar et al. 2023) in real-world tasks, so we have made it available in the package as a stand-alone algorithm.

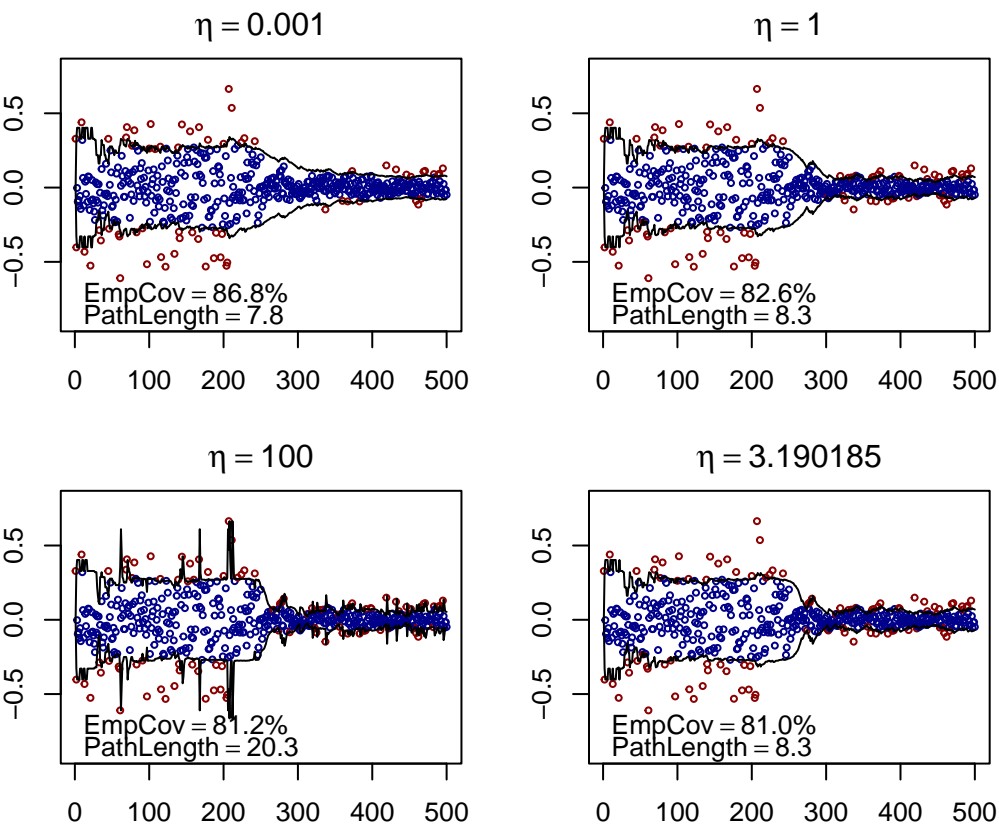

Figure 3: Example 80% prediction intervals generated by the DtACI algorithm with starting values $\theta_1 = 0.8$ and with several values of the tuning parameter $\eta$. Blue and red points are observations that fell inside and outside the prediction intervals, respectively.

**Algorithm 4** Scale-Free Online Gradient Descent

---

1: **Input:** starting value $\theta_1$, learning rate $\gamma > 0$.
2: **for** $t = 1, 2, \dots, T$ **do**
3:   **Output:** prediction interval $\hat{C}_t(\theta_t)$.
4:   Observe $y_t$ and compute $r_t$.
5:   Update $\theta_{t+1} = \theta_t - \gamma \dfrac{\nabla L^\alpha(\theta_t, r_t)}{\sqrt{\sum_{i=1}^{t} \|\nabla L^\alpha(\theta_i, r_i)\|_2^2}}$.
6: **end for**

---

### 3.4.1 Theoretical Guarantees

The SF-OGD algorithm with linear interval constructor has the following regret bound, which is called an *anytime regret bound* because it holds for all $t \in [\![T]\!]$ (Bhatnagar et al. 2023). For any $\gamma > 0$,

$$\text{Reg}(t) \le \mathcal{O}(D\sqrt{t}) \text{ for all } t \in [\![T]\!].$$

A bound for the coverage error has also been established (Bhatnagar et al. (2023); Theorem 4.2). For any learning rate $\gamma = \Theta(D)$ (where $\gamma = D/\sqrt{3}$ is optimal) and any starting value $\theta_1 \in [0, D]$, then it holds that for any $T > 1$,

$$|\text{CovErr}(T)| \le \mathcal{O}\left((1-\alpha)^{-2} T^{-1/4} \log T\right).$$

### 3.4.2 Tuning parameters

Figure 4 compares results for several choices of $\gamma$ to illustrate its effect. The optimal choice of learning rate is $\gamma = D/\sqrt{3}$, where $D$ is the maximum possible radius. When $D$ is not known, it can be estimated by using an initial subset of the time series as a calibration set and estimating $D$ as the maximum of the absolute residuals of the observations and the predictions (Bhatnagar et al. 2023). Figure 4 illustrates SF-OGD for several values of $\gamma$. In the example, the prediction intervals are not reactive enough and do not achieve optimal coverage when $\gamma$ is small. As $\gamma$ increases, the coverage error is near optimal, although the path length becomes larger.

## 3.5 Strongly Adaptive Online Conformal Prediction (SAOCP)

The Strongly Adaptive Online Conformal Prediction (SAOCP; Algorithm 5 ) algorithm was proposed as an improvement over the extant ACI algorithms in that it features stronger theoretical guarantees. SAOCP works similarly to AgACI and DtACI in that it maintains a library of candidate online learning algorithms that generate prediction intervals which are then aggregated using a meta-algorithm (Bhatnagar et al. 2023). The candidate algorithm was chosen to be SF-OGD, although any algorithm that features anytime regret guarantees can be chosen. As opposed to AgACI and DtACI, in which each candidate has a different learning rate but is always able to contribute to the final prediction intervals, here each candidate has the same learning rate but only has positive weight over a specific interval of time. New candidate algorithms are continually being spawned in order that, if the distribution shifts rapidly, the newer candidates will be able to react quickly and receive positive weight. Specifically, at each time point, a new expert is instantiated which is active over a finite "lifetime". Define the *lifetime* of an expert instantiated at time $t$ as

$$L(t) := g \cdot \max_{n \in \mathbb{Z}}\{2^n t \equiv 0 \mod 2^n\},$$

where $g \in \mathbb{Z}^*$ is a *lifetime multiplier* parameter. The active experts are weighted according to their empirical performance with respect to the pinball loss function. The authors show that this construction results in intervals that have strong regret guarantees. The form of the lifetime interval function $L(t)$ is due to the use of geometric covering intervals to partition the input time series, and other choices may be possible (Jun et al. 2017).

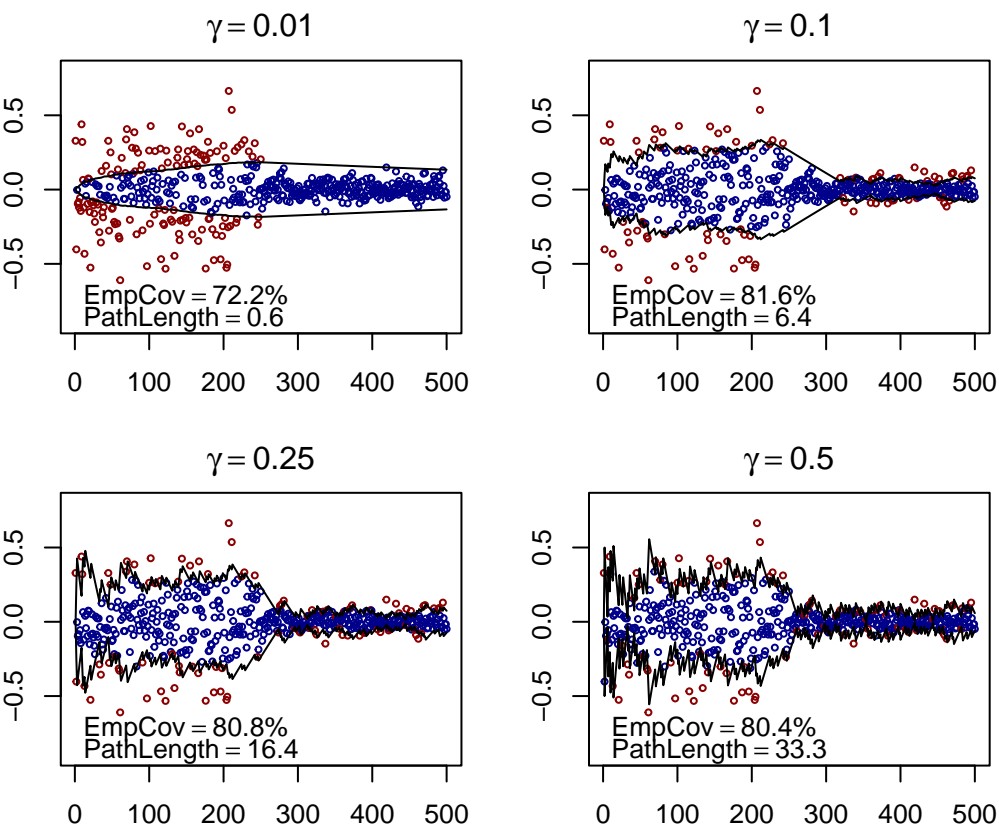

Figure 4: Example 80% prediction intervals generated by the SF-OGD algorithm with different values of the maximum radius tuning parameter $D$. Blue and red points are observations that fell inside and outside the prediction intervals, respectively.

**Algorithm 5** Strongly Adaptive Online Conformal Prediction

1: **Input:** initial value $\theta_0$, learning rate $\gamma > 0$.
2: **for** $t = 1, 2, \ldots, T$ **do**
3:     Initialize expert $\mathscr{A}_t = \text{SF-OGD}(\alpha \leftarrow \alpha, \gamma \leftarrow \gamma, \theta_1 \leftarrow \theta_{t-1})$, set weight $p_t^t = 0$.
4:     Compute active set $\text{Active}(t) = \{i \in [\![T]\!] : t - L(i) < i \leq t\}$ (see below for definition of $L(t)$).
5:     Compute prior probability $\pi_i \propto i^{-2}(1 + \lfloor \log_2 i \rfloor)^{-1} \mathbb{I}[i \in \text{Active}(t)]$.
6:     Compute un-normalized probability $\hat{p}_i = \pi_i [p_{t,i}]_+$ for all $i \in [\![t]\!]$.
7:     Normalize $p = \hat{p}/\|\hat{p}\|_1 \in \Delta^t$ if $\|\hat{p}\|_1 > 0$, else $p = \pi$.
8:     Set $\theta_t = \sum_{i \in \text{Active}(t)} p_i \theta_t^i$ (for $t \geq 2$), and $\theta_t = 0$ for $t = 1$.
9:     **Output:** prediction set $\hat{C}_t(\theta_t)$.
10:     Observe $y_t$ and compute $r_t$.
11:     **for** $i \in \text{Active}(t)$ **do**
12:         Update expert $\mathscr{A}_t$ with $y_t$ and obtain $\theta_{t+1}^i$.
13:         Compute $g_t^i = \begin{cases} \frac{1}{D}\left(L^\alpha(\theta_t, r_t) - L^\alpha(\theta_t^i, r_t)\right) & p_t^i > 0 \\ \frac{1}{D}\left[L^\alpha(\theta_t, r_t) - L^\alpha(\theta_t^i, r_t)\right]_+ & p_t^i \leq 0 \end{cases}$.
14:         Update expert weight $p_{t+1}^i = \frac{1}{t-i+1}\left(\sum_{j=i}^t g_j^i\right)\left(1 + \sum_{j=i}^t p_j^i g_j^i\right)$.
15:     **end for**
16: **end for**

### 3.5.1  Theoretical Guarantees

The theoretical results were established for SAOCP using the linear interval constructor. The following bound for the strongly adaptive regret holds for all subperiod lengths $m \in [\![T]\!]$ (Bhatnagar et al. (2023); Proposition 4.1):

$$\text{SAReg}(T, m) \leq 15D\sqrt{m(\log T + 1)} \leq \tilde{\mathcal{O}}(D\sqrt{m}).$$

It should be emphasized that this regret bounds holds simultaneously across all $m$, as opposed to DtACI, where a similar bound holds only for a single $m$. A bound on the coverage error of SAOCP has also been established as:

$$|\text{CovErr}(T)| \leq \mathcal{O}\left(\inf_\beta(T^{1/2-\beta} + T^{\beta-1}S_\beta(T))\right).$$

where $S_\beta(T)$ is a technical measure of the smoothness of the cumulative gradients and expert weights for each of the candidate experts (Bhatnagar et al. (2023); Theorem 4.3). For some intuition, $S_\beta$ can be expected to be small when the weights placed on each algorithm do change quickly, as would be the case under abrupt distributional shifts.

### 3.5.2  Tuning Parameters

The primary tuning parameter for SAOCP is the learning rate $\gamma$ of the SF-OGD sub-algorithms, which we saw in the previous section has for optimal choice $\gamma = D/\sqrt{3}$. Values for $D$ that are too low lead to intervals that adapt slowly, and values that are too large lead to jagged intervals. In their experiments, the authors select a value for $D$ by picking the maximum residual from a calibration set. The second tuning parameter is the lifetime multiplier $g$ which controls the lifetime of each of the experts. We follow the original paper in setting $g = 8$. Figure 5 illustrates the SAOCP algorithm for choices of $D \in \{0.01, 0.1, 0.25, 0.5\}$. Similarly to SF-OGD, the prediction intervals tend to undercover for small $D$, and achieve near-optimal coverage for larger $D$ at the expense of larger path lengths.

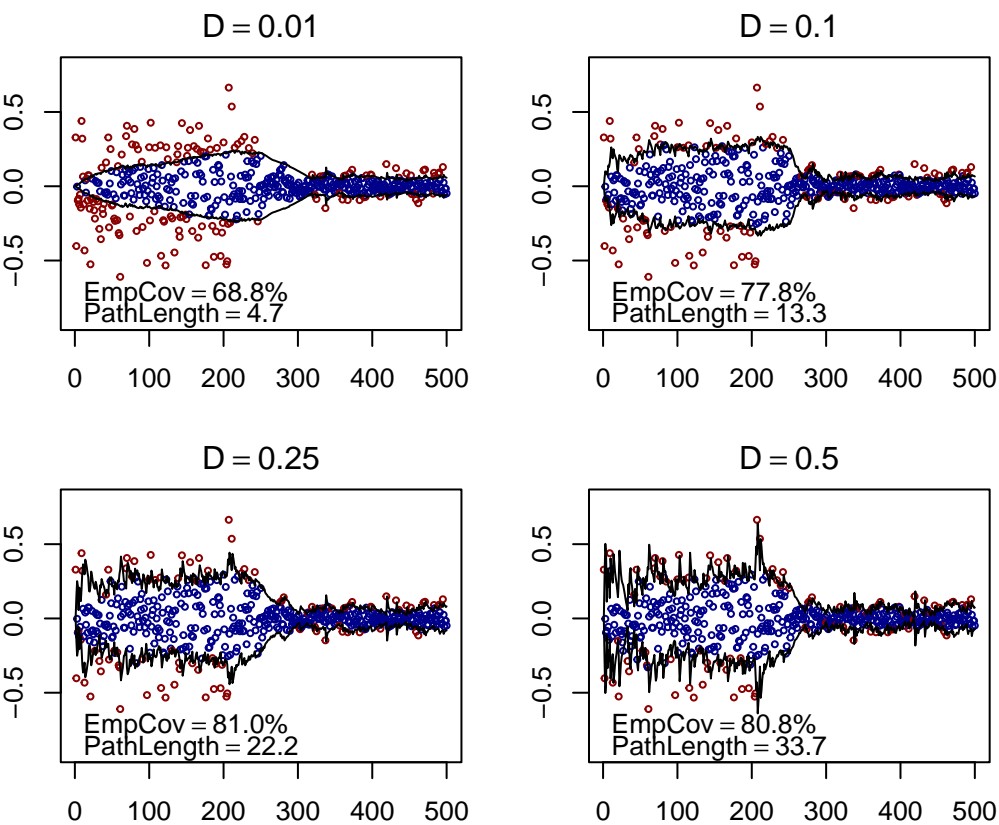

Figure 5: Example 80% prediction intervals generated by the SAOCP algorithm with different values of the maximum radius parameter $D$. Blue and red points are observations that fell inside and outside the prediction intervals, respectively.

# 4  `AdaptiveConformal` R **package**

The ACI algorithms described in the previous section have been implemented in the open-source and publically available R package `AdaptiveConformal`, available at https://github.com/herbps10/AdaptiveConformal. CIn this section, we briefly introduce the main functionality of the package. Comprehensive documentation is, including several example vignettes, is included with the package.

The `AdaptiveConformal` package can be installed using the `remotes` package:

```r
remotes::install_github("herbps10/AdaptiveConformal")
```

The ACI algorithms are accessed through the `aci` function, which takes as input a vector of observations ($y_t$) and a vector or matrix of predictions ($\hat{y}_t$). Using the data generating process from the running example to illustrate, we can fit the original ACI algorithm with learning rate $\gamma = 0.1$:

```r
set.seed(532)
data <- running_example_data(N = 5e2)
fit <- aci(data$y, data$yhat, alpha = 0.8, method = "ACI", parameters = list(gamma = 0.1))
```

The available parameters for each method can be found in the documentation for the `aci` method, accessible with the command `?aci`. The resulting conformal prediction intervals can then be plotted using the `plot` function:

```r
plot(fit)
```

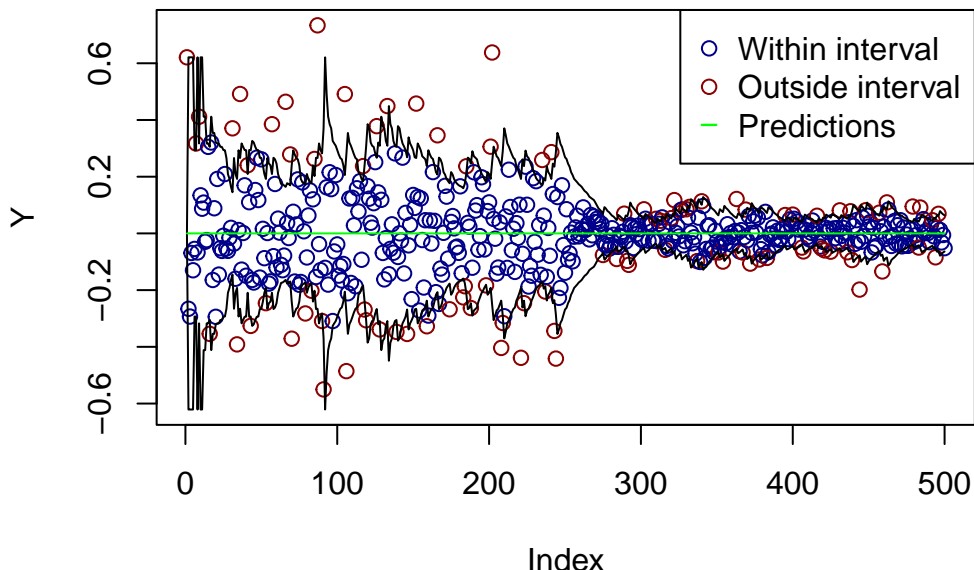

The properties of the prediction intervals can also be examined using the `summary` function:

```r
summary(fit)
```

```
Method: ACI
Empirical coverage: 80.6% (403/500)
Below interval: 10.2%
Above interval: 9.2%
```

```
Mean interval width: 0.354
Mean interval loss: 0.498
```

# 5   Simulation Studies

We present two empirical studies in order to compare the performance of the AgACI, DtACI, SF-OGD, and SAOCP algorithms applied to simple simulated datasets. The original ACI algorithm was not included as it is not clear how to set the tuning rate $\gamma$, which can have a large effect on the resulting intervals. For both simulations we set the targeted empirical coverage to $\alpha = 0.8$, $\alpha = 0.9$, and $\alpha = 0.95$. For each algorithm, we chose the interval constructor that was used in its original presentation (see Table 1).

## 5.1   Time series with ARMA errors

In this simulation we reproduce the setup described in Zaffran et al. (2022) (itself based on that of Friedman, Grosse, and Stuetzle (1983)). The time series values $y_t$ for $t \in [\![T]\!]$ ($T = 600$) are simulated according to

$$y_t = 10 \sin(\pi X_{t,1} X_{t,2}) + 20(X_{t,3} - 0.5)^2 + 10 X_{t,4} + 5 X_{t,5} + 0 X_{t,6} + \epsilon_t,$$

where $X_{t,i}$, $i = 1, \ldots, 6$, $t \in [\![T]\!]$ are independently uniformly distributed on $[0, 1]$ and the noise terms $\epsilon_t$ are generated according to an ARMA(1, 1) process:

$$\epsilon_t = \psi \epsilon_{t-1} + \xi_t + \theta \xi_{t-1},$$
$$\xi_t \sim N(0, \sigma^2).$$

We set $\psi$ and $\theta$ jointly to each value in $\{0.1, 0.8, 0.9, 0.95, 0.99\}$ to simulate time series with increasing temporal dependence. The innovation variance was set to $\sigma^2 = (1 - \psi^2)/(1 + 2\psi\xi + \xi^2)$ (to ensure that the process has constant variance). For each setting, 25 simulated datasets were generated.

To provide point predictions for the ACI algorithms, at each time $t \geq 200$ a random forest model was fitted to the previously observed data using the `ranger` R package (Wright and Ziegler 2017). The estimated model was then used to predict the subsequent time point. The maximum radius $D$ was estimated as the maximum residual observed between time points $t = 200$ and $t = 249$. The ACI models were then executed starting at time point $t = 250$. All metrics are based on time points $t \geq 300$ to allow time for the ACI methods to initialize.

```
simulate <- function(seed, psi, xi, N = 1e3) {
  set.seed(seed)

  s <- 10
  innov_scale <- sqrt(s * (1 - psi^2) / (1 + 2 * psi * xi + xi^2))

  X <- matrix(runif(6 * N), ncol = 6, nrow = N)
  colnames(X) <- paste0("X", 1:6)

  epsilon <- arima.sim(n = N, model = list(ar = psi, ma = xi), sd = innov_scale)

  mu <- 10 * sin(pi * X[,1] * X[,2]) + 20 * (X[,3] - 0.5)^2 + 10 * X[,4] + 5 * X[,5]
  y <- mu + epsilon
  as_tibble(X) %>% mutate(y = y)
```

```r
}

estimate_model <- function(data, p = NULL) {
  if(!is.null(p)) p()
  preds <- numeric(nrow(data))
  for(t in 200:nrow(data)) {
    model <- ranger::ranger(y ~ X1 + X2 + X3 + X4 + X5 + X6, data = data[1:(t - 1),])
    preds[t] <- predict(model, data = data[t, ])$predictions
  }
  preds
}

metrics <- function(fit) {
  indices <- 300:length(fit$Y)
  aci_metrics(fit, indices)
}

fit <- function(data, preds, method, alpha, p = NULL) {
  if(!is.null(p)) p()

  D <- max(abs(data$y - preds)[200:249])
  gamma <- D / sqrt(3)

  interval_constructor = case_when(
    method == "AgACI" ~ "conformal",
    method == "DtACI" ~ "conformal",
    method == "SF-OGD" ~ "linear",
    method == "SAOCP" ~ "linear"
  )

  if(interval_constructor == "linear") {
    gamma_grid = seq(0.1, 1, 0.1)
  }
  else {
    gamma_grid <- c(0.001, 0.002, 0.004, 0.008, 0.016, 0.032, 0.064, 0.128)
  }

  parameters <- list(
    interval_constructor = interval_constructor,
    D = D,
    gamma = gamma,
    gamma_grid = gamma_grid
  )

  aci(
    data$y[250:nrow(data)],
    preds[250:nrow(data)],
    method = method,
    alpha = alpha,
```

```
      parameters = parameters
  )
}

N_sims <- 100
simulation_data <- expand_grid(
  index = 1:N_sims,
  param =  c(0.1, 0.8, 0.9, 0.95, 0.99),
  N = 600
) %>%
  mutate(psi = param, xi = param)

# For each simulated dataset, fit multiple ACI methods
simulation_study_setup <- expand_grid(
  alpha = c(0.8, 0.9, 0.95),
  method = c("AgACI", "SF-OGD", "SAOCP", "DtACI")
)

# run_simulation_study1 function is defined in helpers.R
simulation_study1 <- run_simulation_study1(
  simulation_data,
  simulation_study_setup,
  estimate_model,
  fit,
  workers = 8
)
```

The coverage errors, mean interval widths, path lengths, and strongly adaptive regret (for $m = 20$) of each of the algorithms for $\alpha = 0.9$ are shown in Figure 6 (results for $\alpha \in \{0.8, 0.95\}$ were similar and are available in the appendix). All methods achieved near optimal empirical coverage, although SAOCP tended to slightly undercover. The mean interval widths re similar across methods, although again SAOCP had slightly shorter intervals (as could be expected given its tendency to undercover). The strongly adaptive regret was similar for all methods. The path length of SAOCP was larger than any of the other methods. To investigate why, Figure 7 plots $w_t - w_{t-1}$, the difference in interval width between times $t - 1$ and $t$, for each method in one of the simulations. The interval widths for AgACI and DtACI change slowly relative to those for SF-OGD and SAOCP. For SAOCP, we can see the interval widths have larger fluctuations than for the other methods, explaining its higher path width. The prediction intervals themselves for the same simulation are shown in Figure 8, which shows that although the path lengths are quite different, the output prediction intervals are quite similar.

```
simulation_one_plot(simulation_study1$results %>% filter(alpha == 0.9))
```

Figure 6: Coverage errors, mean interval widths, path lengths, and strongly adaptive regret (for $m = 20$) for the first simulation study with target coverage $\alpha = 0.9$.

```r
fits <- simulation_study1$example_fits

par(mfrow = c(2, 2), mar = c(3, 4, 2, 1))
for(i in 1:4) {
  plot(
    diff(fits$fit[[i]]$intervals[,2] - fits$fit[[i]]$intervals[,1]),
    main = fits$method[[i]],
    xlab = "T",
    ylab = expression(w[t] - w[t - 1]))
}
par(mfrow = c(1, 1), mar = c(5.1, 4.1, 4.1, 2.1))
```

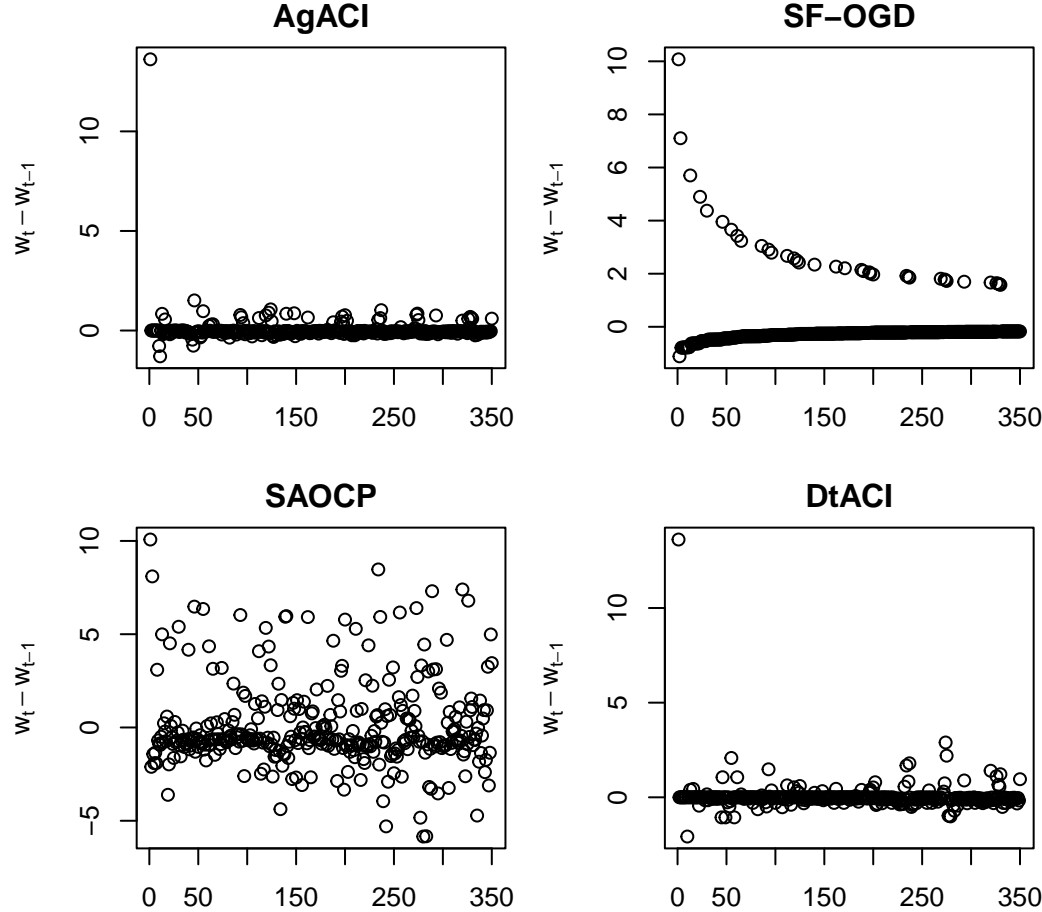

Figure 7: Difference in successive interval widths ($w_t - w_{t-1}$) from an illustrative simulation from the first simulation study.

```r
fits <- simulation_study1$example_fits

coverage    <- format_coverage(map_dbl(map(fits$fit, metrics), `[[`, "coverage"))
path_length <- format_path_length(map_dbl(map(fits$fit, metrics), `[[`, "path_length"))

par(mfrow = c(2, 2), mar = c(3, 3, 2, 1))
for(i in 1:4) {
```

```
    plot(fits$fit[[i]], legend = FALSE, main = fits$method[[i]], predictions = FALSE, ylim = c(-20
    text(x = -0, y = -7.5, labels = bquote(EmpCov == .(coverage[[i]]) ), pos = 4)
    text(x = -0, y = -17.5, labels = bquote(PathLength == .(path_length[[i]]) ), pos = 4)
  }
  par(mfrow = c(1, 1), mar = c(5.1, 4.1, 4.1, 2.1))
```

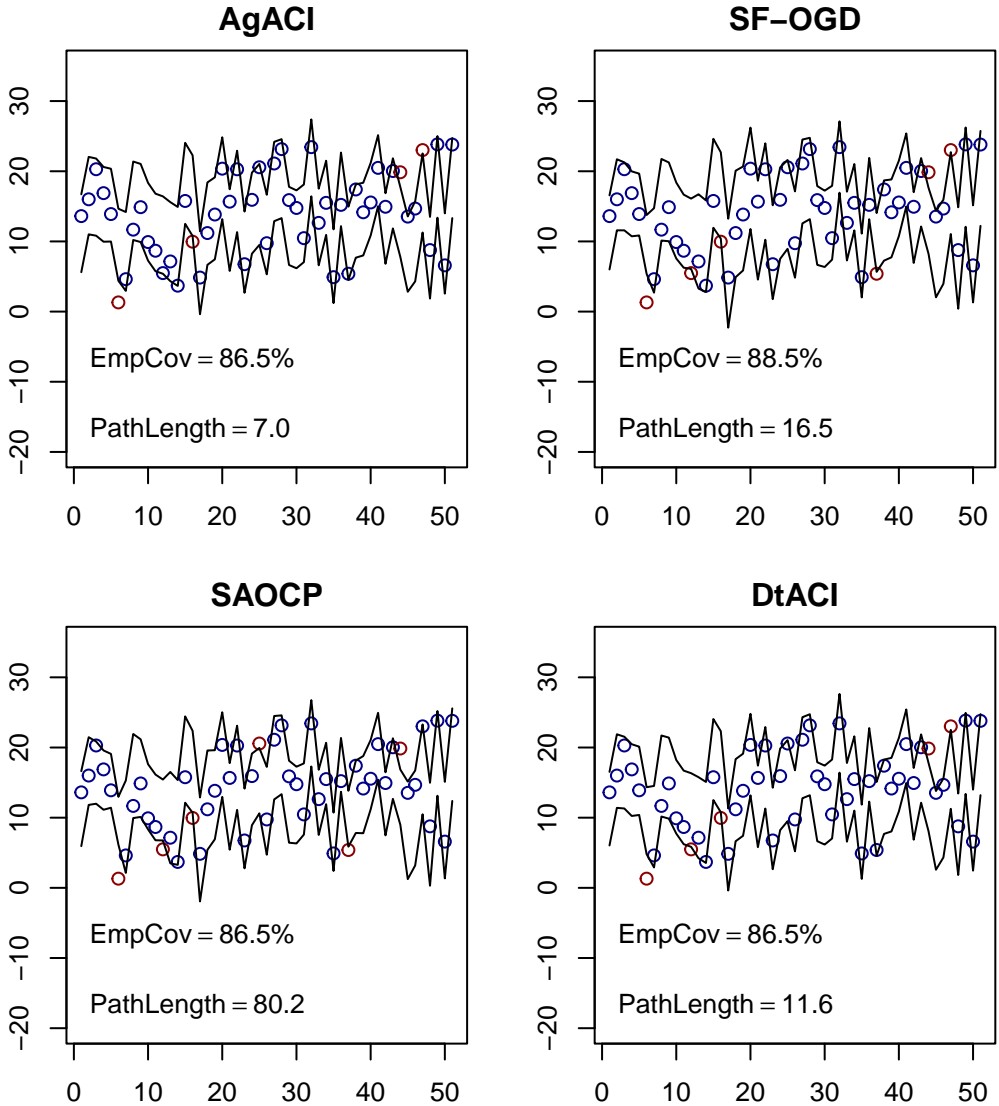

Figure 8: Example prediction intervals (target coverage $\alpha = 0.9$) from the first simulation study for time points 300 to 350; metrics shown are for all time points $t \geq 300$. Blue and red points are observations that fell inside and outside the prediction intervals, respectively.

## 5.2 Distribution shift

This simulation study features time series with distribution shifts. The setup is quite simple in order to probe the basic performance of the methods in response to distribution shift. As a baseline, we simulate time series of independent data with

$$y_t \sim N(0, \sigma_t^2),$$
$$\sigma_t = 0.2,$$

for all $t \in [\![T]\!]$ ($T = 500$). In the second type of time series, the observations are still independent but their variance increases halfway through the time series:

$$y_t \sim N(0, \sigma_t^2),$$
$$\sigma_t = 0.2 + 0.5\mathbb{I}[t > 250].$$

In each case, the ACI algorithms are provided with the unbiased predictions $\hat{\mu}_t = 0, t \in [\![T]\!]$. Fifty simulated datasets were generated for each type of time series.

```r
simulate <- function(seed, distribution_shift = 0, N = 1e3, sigma = 0.2) {
  set.seed(seed)
  mu <- rep(0, N)
  shift <- 1:N > (N / 2)
  yhat <- mu
  y <- rnorm(n = length(mu), mean = mu, sd = sigma + ifelse(shift, distribution_shift, 0))

  tibble(y = y, yhat = yhat)
}

metrics <- function(fit) {
  N <- length(fit$Y)
  indices <- which(1:N > 50)
  aci_metrics(fit, indices)
}

fit <- function(data, method, alpha, p = NULL) {
  if(!is.null(p)) p()

  interval_constructor = case_when(
    method == "AgACI" ~ "conformal",
    method == "DtACI" ~ "conformal",
    method == "SF-OGD" ~ "linear",
    method == "SAOCP" ~ "linear"
  )

  if(interval_constructor == "linear") {
    D <- max(abs(data$y - data$yhat)[1:50])
  }
  else {
    D <- 1
  }

  gamma <- D / sqrt(3)

  if(interval_constructor == "linear") {
    gamma_grid <- seq(0.1, 2, 0.1)
  }
  else {
    gamma_grid <- c(0.001, 0.002, 0.004, 0.008, 0.016, 0.032, 0.064, 0.128)
  }
```

```
  parameters <- list(
    interval_constructor = interval_constructor,
    D = D,
    gamma = gamma,
    gamma_grid = gamma_grid
  )

  aci(data$y, data$yhat, method = method, alpha = alpha, parameters = parameters)
}

N_sims <- 100
simulation_study_setup2 <- expand_grid(
  index = 1:N_sims,
  distribution_shift = c(0, 0.5),
  alpha = c(0.8, 0.9, 0.95),
  N = 500,
  method = c("AgACI", "SF-OGD", "SAOCP", "DtACI"),
) %>%
  mutate(data = pmap(list(index, distribution_shift, N), simulate))

# run_simulation_study2 function is defined in helpers.R
simulation_study2 <- run_simulation_study2(simulation_study_setup2, fit, workers = 8)
```

The coverage error, mean path length, mean interval widths, and strongly adaptive regret (for $m = 20$) of the algorithms are summarized in Figure 9 (an alternative plot is included in the appendix as Figure 15). The coverage error of all the algorithms is near the desired value in the absence of distribution shift. On the contrary, all of the algorithms except AgACI and DtACI undercover when there is distributional shift. SAOCP tends to have higher average path lengths than the other methods. In the distribution shift setting, SF-OGD and SAOCP tended to have smaller strongly adaptive regret than the other methods. An illustrative example of prediction intervals generated by each method for one of the simulated time series with distribution shift is shown in Figure 10. The SAOCP prediction intervals in the example before the distribution shift are more jagged than those produced by the other methods, which illustrates why SAOCP may have longer path lengths.

```
simulation_two_plot(simulation_study2$results)
```

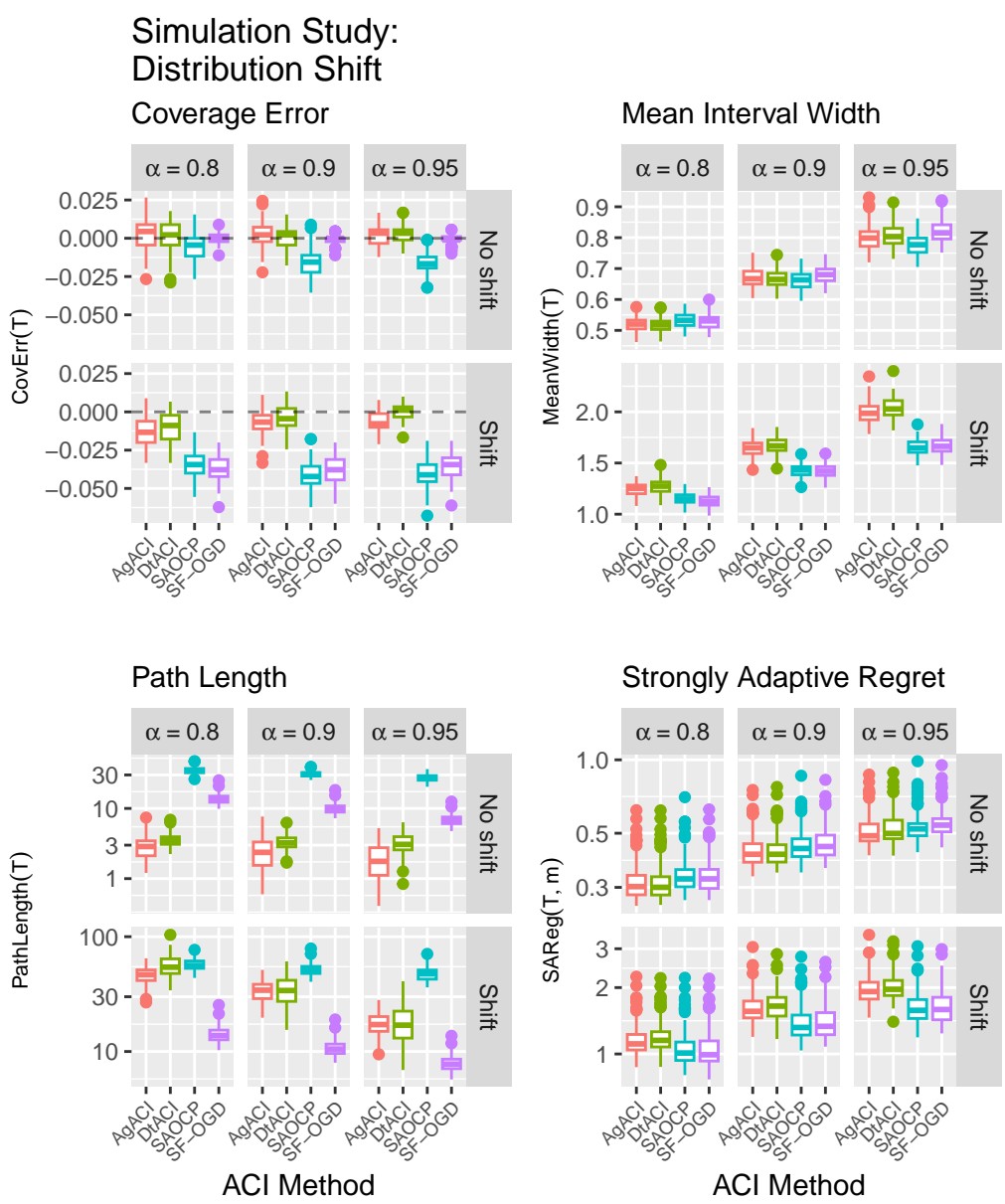

Figure 9: Coverage error, mean interval width, path length, and strongly adaptive regret ($m = 20$) for $\alpha = 0.8, 0.9, 0.95$ and simulations with and without distributional shift.

```r
fits <- simulation_study2$example_fits

coverage    <- format_coverage(extract_metric(fits$fit, "coverage"))
path_length <- format_path_length(extract_metric(fits$fit, "path_length"))

par(mfrow = c(2, 2), mar = c(3, 3, 2, 1))
for(i in 1:4) {
  plot(fits$fit[[i]], legend = FALSE, main = fits$method[[i]], index = 51:500)
  text(x = -10, y = -1.5, labels = bquote(EmpCov == .(coverage[[i]]) ), pos = 4)
  text(x = -10, y = -2, labels = bquote(PathLength == .(path_length[[i]]) ), pos = 4)
}
par(mfrow = c(1, 1), mar = c(5.1, 4.1, 4.1, 2.1))
```

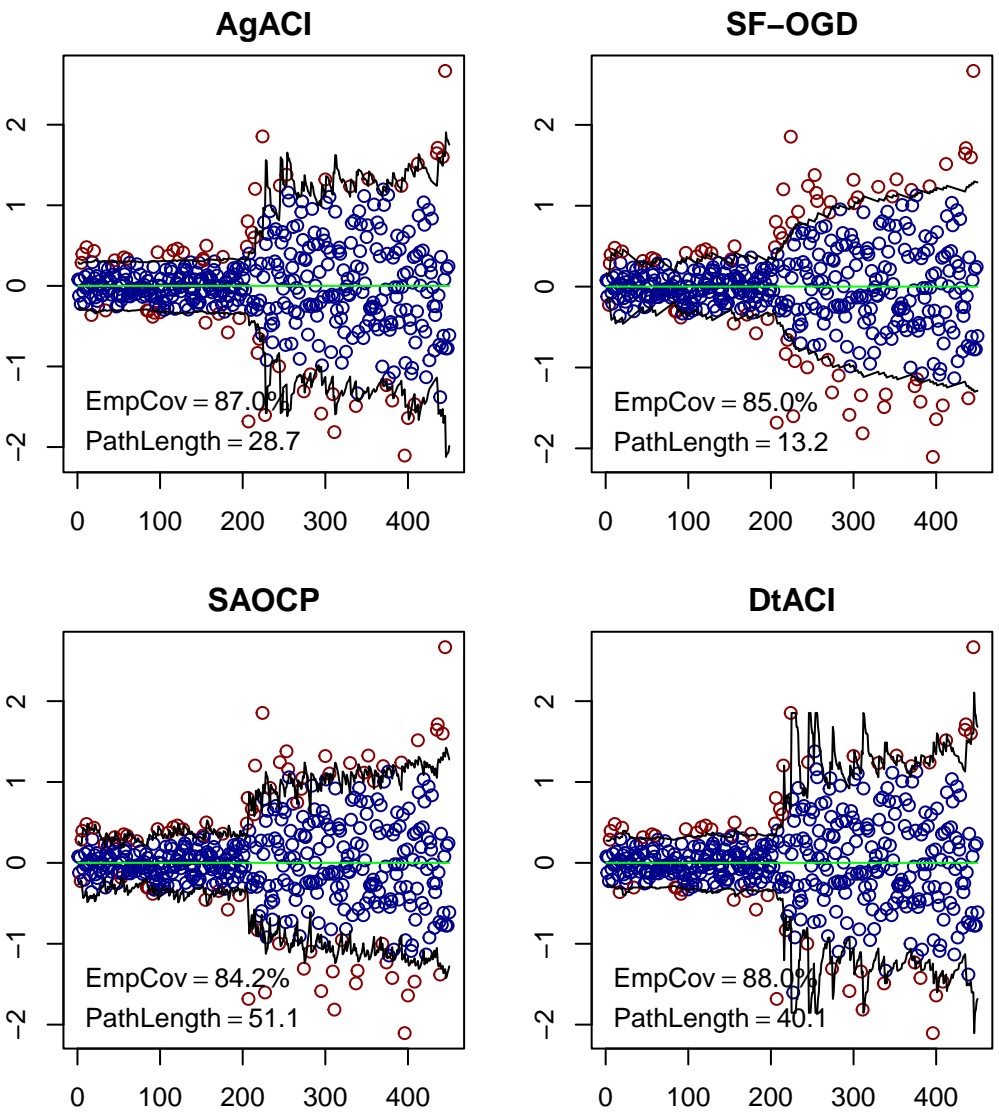

Figure 10: Example prediction intervals (target coverage $\alpha = 0.9$) from the second simulation study of time series with distributional shift, in which the shift occurs at time 250. Blue and red points are observations that fell inside and outside the prediction intervals, respectively.

## 6 Case Study: Influenza Forecasting

Influenza is a highly infectious disease that is estimated to infect approximately one billion individuals each year around the world (Krammer et al. 2018). Influenza incidence in temperate climates tends to follow a seasonal pattern, with the highest number of infections during what is commonly referred to as the *flu season* (Lofgren et al. 2007). Accurate forecasting of influenza is of significant interest to aid in public health planning and resource allocation. To investigate the accuracy of influenza forecasts, the US Centers for Disease Control (CDC) initiated a challenge, referred to as FluSight, in which teams from multiple institutions submitted weekly forecasts of influenza incidence (Biggerstaff et al. 2016). Reich et al. (2019) evaluated the accuracy of the forecasts over seven flu seasons from 2010 to 2017. As a case study, we investigate the use of ACI algorithms to augment the FluSight forecasts with prediction intervals.

The FluSight challenge collected forecasts for multiple prediction targets. For this case study, we focus on national (US) one-week ahead forecasts of weighted influenza-like illness (wILI), which

is a population-weighted percentage of doctors visits where patients presented with influenza-like symptoms (Biggerstaff et al. 2016). The FluSight dataset, which is publicly available, include forecasts derived from 21 different forecasting models, from both mechanistic and statistical viewpoints (Flusight Network 2020; Tushar et al. 2018, 2019). For our purposes, we treat the way the forecasts were produced as a black box.

Formally, let $y_t$, $t \in [\![T]\!]$ be the observed national wILI at time $t$, and let $\hat{\mu}_{j,t}$, $j \in [\![J]\!]$, be the one-week ahead forecast of the wILI from model $j$ at time $t$. Two of the original 21 forecasting methods were excluded from this case study due to poor predictive performance (`Delphi_Uniform` and `CUBMA`). In addition, six methods had identical forecasts (`CU_EAKFC_SIRS`, `CU_EKF_SEIRS`, `CU_EKF_SIRS`, `CU_RHF_SEIRS`, `CU_RHF_SIRS`), and therefore we only included one (`CU_EAKFC_SIRS`) in the analysis. The ACI methods were then applied to the log-observations and log-predictions, where the log-transformation was used to constrain the final prediction intervals to be positive. The first flu season (2010-2011) was used as a warm-up for each ACI method, and we report the empirical performance of the prediction intervals for the subsequent seasons (six seasons from 2012-2013 to 2016-2017). The ACI algorithms target prediction intervals with coverage of $\alpha = 0.8$, $\alpha = 0.9$, and $\alpha = 0.95$. As in the simulation study, we used the interval constructor corresponding to the original presentation of each algorithm (see Table 1).

```r
# Paste together URL so it is not cut off in PDF
url <- paste0("https://raw.githubusercontent.com/FluSightNetwork/",
 "cdc-flusight-ensemble/master/scores/point_ests.csv")
raw_data <- read_csv(url, show_col_types = FALSE)

fit <- function(data, method, alpha) {
  first_season <- data$Season == "2010/2011"
  D <- max(abs(data$obs_value - data$Value)[first_season])

  interval_constructor = case_when(
    method == "AgACI" ~ "conformal",
    method == "DtACI" ~ "conformal",
    method == "SF-OGD" ~ "linear",
    method == "SAOCP" ~ "linear"
  )

  gamma <- D / sqrt(3)

  if(interval_constructor == "linear") {
    gamma_grid = seq(0.1, 1, 0.1)
  }
  else {
    gamma_grid <- c(0.001, 0.002, 0.004, 0.008, 0.016, 0.032, 0.064, 0.128)
  }

  parameters <- list(
    interval_constructor = interval_constructor,
    D = D,
    gamma = gamma,
    gamma_grid = gamma_grid
  )
```

```
  aci(
    Y = log(data$obs_value),
    predictions = log(data$Value),
    method = method,
    parameters = parameters,
    alpha = alpha
  )
}

metrics <- function(data, fit) {
  aci_metrics(fit, indices = which(data$Season != "2010/2011"))
}

analysis_data <- raw_data %>%
  filter(
    Target == "1 wk ahead",
    Location == "US National",
    !(model_name %in% c("Delphi_Uniform", "CUBMA", "CU_EAKFC_SIRS", "CU_EKF_SEIRS",
                        "CU_EKF_SIRS", "CU_RHF_SEIRS", "CU_RHF_SIRS"))
  ) %>%
  arrange(Year, Model.Week) %>%
  group_by(model_name) %>%
  nest()

fits <- expand_grid(
  analysis_data,
  tibble(method = c("AgACI", "DtACI", "SF-OGD", "SAOCP")),
  tibble(alpha = c(0.8, 0.9, 0.95))
) %>%
  mutate(fit = pmap(list(data, method, alpha), fit),
         metrics = map2(data, fit, metrics))

case_study_results <- fits %>%
  select(-data, -fit) %>%
  mutate(metrics = map(metrics, as_tibble)) %>%
  unnest(c(metrics))
```

The coverage errors, mean interval widths, path lengths, and strongly adaptive regret (for $m = 20$) of the prediction intervals for each of the underlying forecast models is shown in Figure 11. In all cases the absolute coverage error was less than 0.1. SF-OGD performed particularly well, with coverage errors close to zero for all forecasting models. Interval widths were similar across methods, with SAOCP slightly shorter. Path Lengths were shorter for AgACI and DtACI and longer for SAOCP.

```
  case_study_plot(case_study_results)
```

Figure 11: Coverage errors, mean interval widths, path lengths, and strongly adaptive regret (for $m = 20$) of prediction intervals generated with each ACI method based on forecasts from each of the 19 underlying influenza forecasting models.

As an illustrative example, in Figure 12 we plot the point forecasts from one of the forecasting models

(based on SARIMA with no seasonal differencing) and the associated ACI-generated 90% prediction intervals for each season from 2011-2017. In general, in this practical setting all of the ACI algorithms yield quite similar prediction intervals. Interestingly, the forecasts in 2011-2012 underpredicted the observations for much of the season. The algorithm responds by making the intervals wider to cover the observations, and because the intervals are symmetric the lower bound then becomes unrealistically low. A similar phenomenon can be seen in the growth phase of the 2012/2013 season as well.

```
sarima_fits <- fits %>% filter(
  model_name == "ReichLab_sarima_seasonal_difference_FALSE",
  alpha == 0.9
) %>%
  mutate(output = map(fit, extract_intervals)) %>%
  select(method, alpha, data, output) %>%
  unnest(c(data, output)) %>%
  filter(Season != "2010/2011")

sarima_fits %>%
  ggplot(aes(x = Model.Week, y = log(obs_value))) +
  geom_point(aes(shape = "Observed")) +
  geom_line(aes(y = pred, lty = "Forecast"), color = "black") +
  geom_line(aes(y = lower, color = method)) +
  geom_line(aes(y = upper, color = method)) +
  facet_wrap(~Season) +
  labs(
    x = "Flu Season Week",
    y = "log(wILI)",
    title = "SARIMA forecasts with ACI 90% prediction intervals"
  )
```

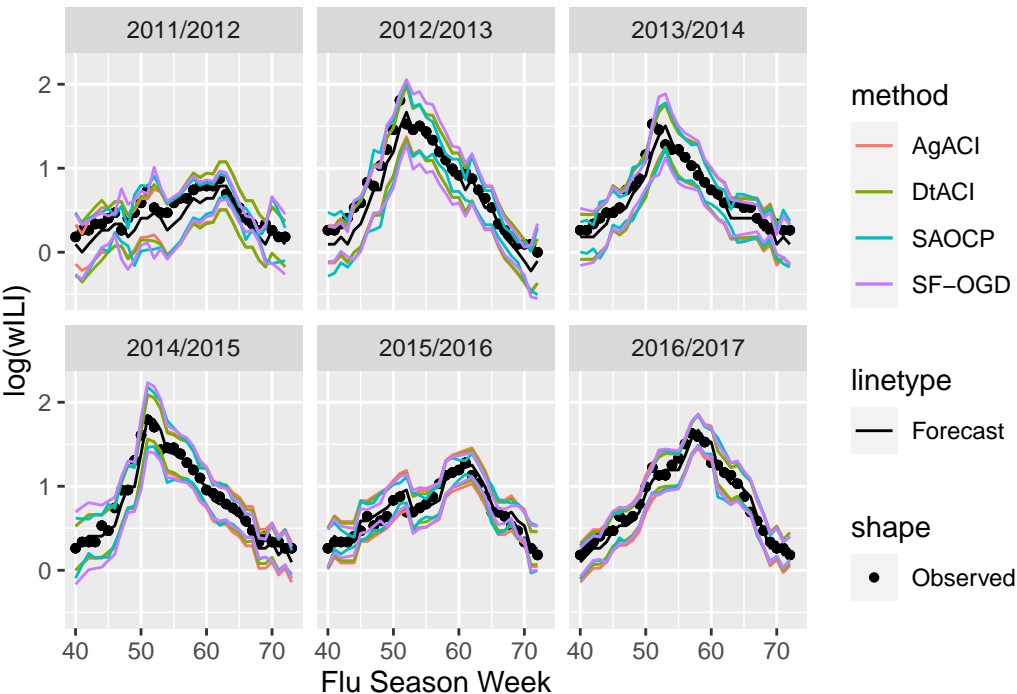

Figure 12: Example conformal prediction intervals for six flu seasons based on forecasts from a SARIMA type model.

## 7 Discussion

The results of our simulations and case study show that, when tuning parameters are chosen well, Adaptive Conformal Inference algorithms yield well-performing prediction intervals. On the contrary, poor choice of tuning parameters can lead to intervals of low utility: for one example, Figure Figure 4 shows how choosing the tuning parameter for SF-OGD to be too small can lead to intervals that update too slowly and significantly undercover. Furthermore, in some cases the prediction intervals may appear to perform well with respect to metrics like the empirical coverage error, while simultaneously being useless in practice. The original ACI algorithm illustrates this phenomenon: too small a value of its learning rate $\gamma$ yields prediction intervals that are not reactive enough, while too large a value yields intervals that change too fast. In both cases, the empirical coverage may appear well-calibrated, while the prediction intervals will not be useful. Thus, the core challenge in designing an ACI algorithm is in finding an optimal level of reactivity for the prediction intervals. As users of these algorithms, the challenge is in finding values for the tuning parameters that avoid pathological behaviors.

Several of the algorithms investigated in this paper handle the problem of finding an optimal level of reactivity by aggregating prediction intervals generated by a set of underlying ACI algorithms. Our results show the algorithms can perform well in multiple difficult scenarios. However, the overall effect of these approaches is to shift the problem to a higher level of abstraction: we still need to set tuning parameters that control the amount of reactivity, but do so at a higher level than the original ACI algorithm. It is desirable that these tuning parameters be easily interpretable, with simple strategies available for setting them. An advantage of the SF-OGD and SAOCP algorithms in this respect are that their main tuning parameter, the maximum radius $D$, is easily interpretable as the maximum possible difference between the input predictions and the truth. It is also straightforward to choose this parameter based on a calibration set, although this strategy does not necessarily work

well in cases of distribution shift. We also found that an advantage of the AgACI method is its robustness to the choice of its main tuning parameter, the set of candidate learning rates, in the sense that the grid of candidate learning rates can always be expanded as illustrated in Section 3.2.2.

A key challenge in tuning the algorithms arises in settings of distribution shift, where methods for choosing hyperparameters based on a calibration set from before the distribution shift will likely not perform well. The second simulation study we conducted probed this setting in a simple scenario. We found that several of the methods yielded prediction intervals that had non-optimal empirical coverage. As we picked hyperparameters based on a calibration set formed before the distribution shift, it is not surprising that the resulting tuning parameters are not optimal. This underscores the difficulty in designing ACI algorithms that can adapt to distribution shifts, and in finding robust methods for choosing hyperparameters. In practice, it is possible the second simulation study does not accurately reflect real-world scenarios. Indeed, the benchmarks presented in Bhatnagar et al. (2023) using the datasets from the M4 competition (Makridakis, Spiliotis, and Assimakopoulos 2020), and using point predictions generated by diverse prediction algorithms, found that ACI algorithms exhibited good performance in terms of empirical coverage. Nevertheless, our recommendation for future papers in this line of research is to include simulation studies for simple distributional shift scenarios as a benchmark.

Our case study results illustrate the dependence of the ACI algorithms on having access to high-quality point predictions. If the predictions are biased, for example, then the prediction intervals may be able to achieve optimal coverage at the expense of larger interval widths. This type of underperformance due to biased input predictions can be seen in the 2011-2012 flu season in the case study Figure 12. One way bias can arise in the underlying predictions is due to model misspecification: for example, if a forecast method assumes a time series will evolve according to a particular parametric model that does not accurately capture the true data generating process, then the forecasts may be systematically biased. Using ensemble methods to combine forecasts from several flexible machine learning algorithms is one strategy that can be used to hedge against such model misspecification and improve the quality of forecasts (Makridakis, Spiliotis, and Assimakopoulos 2020).

Overall, our findings illustrate strengths and weaknesses of all the considered algorithms. The original ACI algorithm is appealing in its simplicity, although its performance depends entirely on a good choice of its tuning parameter. AgACI tended to perform well in the simulation studies in terms of coverage error, although it had slightly higher strongly adaptive regret than other algorithms in some settings. However, there are relatively fewer theoretical guarantees available for AgACI than the other methods. DtACI, SF-OGD, and SAOCP all feature strong theoretical results, although they exhibited some differences in the simulation studies, with SF-OGD and SAOCP slightly undercovering in some scenarios. SAOCP also had longer path lengths than other methods in simulations, although in practice in the influenza forecasting task longer path lengths does not seem to effect the plausibility of the prediction intervals the algorithm produces.

There remain many possible extensions of ACI algorithms. The algorithms presented in this work primarily consider symmetric intervals evaluated using the pinball loss function (AgACI can yield asymmetric intervals because the aggregation rule is applied separately to the lower and upper bounds from the underlying experts, but those underlying experts only produce symmetric intervals). A simple extension would switch to using the interval loss function (Gneiting and Raftery 2007), which would allow for asymmetric intervals where two parameters are learned for the upper and lower bounds, respectively. It may also be of interest to generate prediction intervals that have coverage guarantees for arbitrary subsets of observations (for example, we may seek prediction intervals for daily observations that have near optimal coverage for every day of the week, or month of the year), similar to guarantees provided by the MultiValid Prediction method described in (Bastani et al. 2022). Another avenue for theoretical research is to relax the assumption of bounded radii

necessary for the theoretical results of algorithms such as SAOCP.

## Acknowledgements

This research is partially supported by the Agence Nationale de la Recherche as part of the "Investissements d'avenir" program (reference ANR-19-P3IA-0001; PRAIRIE 3IA Institute). We would like to thank Margaux Zaffran for providing helpful comments on the manuscript.

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

# 8 Appendix

## 8.1 Additional simulation study results

```
simulation_one_plot(simulation_study1$results)
```

# Simulation study: ARMA errors

## Coverage Error

## Mean Interval Width

## Path Length

## Strongly Adaptive Regret

Figure 13: Coverage errors, mean interval widths, and path lengths for the first simulation study with target coverage $\alpha \in \{0.8, 0.9, 0.95\}$.

```
simulation_one_joint_plot(simulation_study1$results)
```

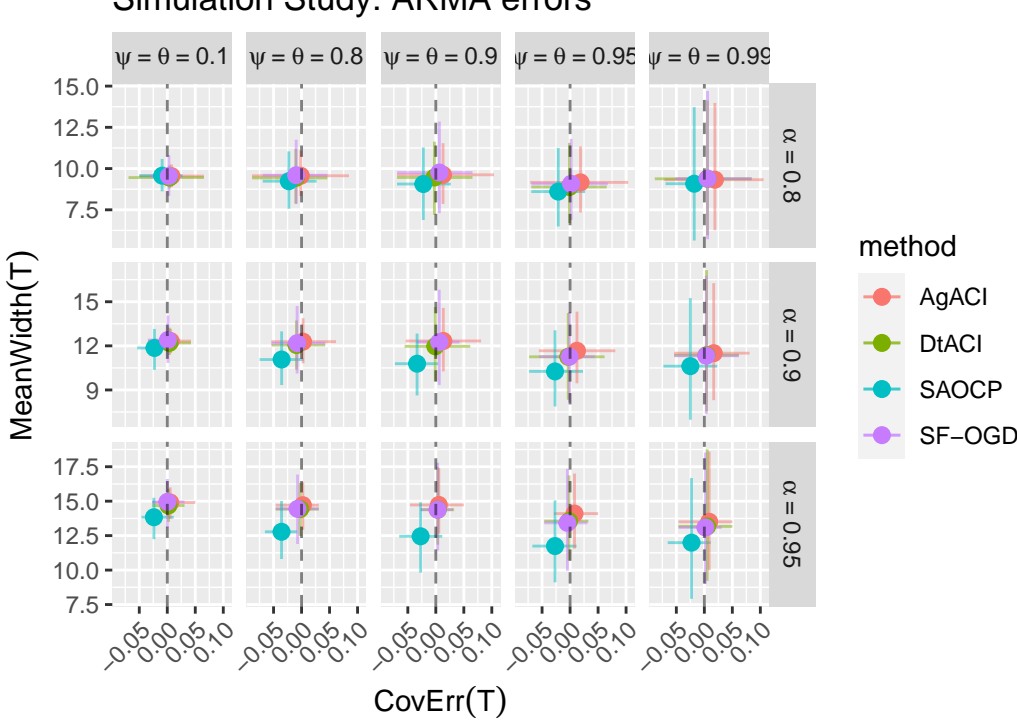

Figure 14: Mean Interval Width vs Coverage Error for the first simulation study. The error bars represent the 10% to 90% quantiles of the metrics over the simulation datasets.

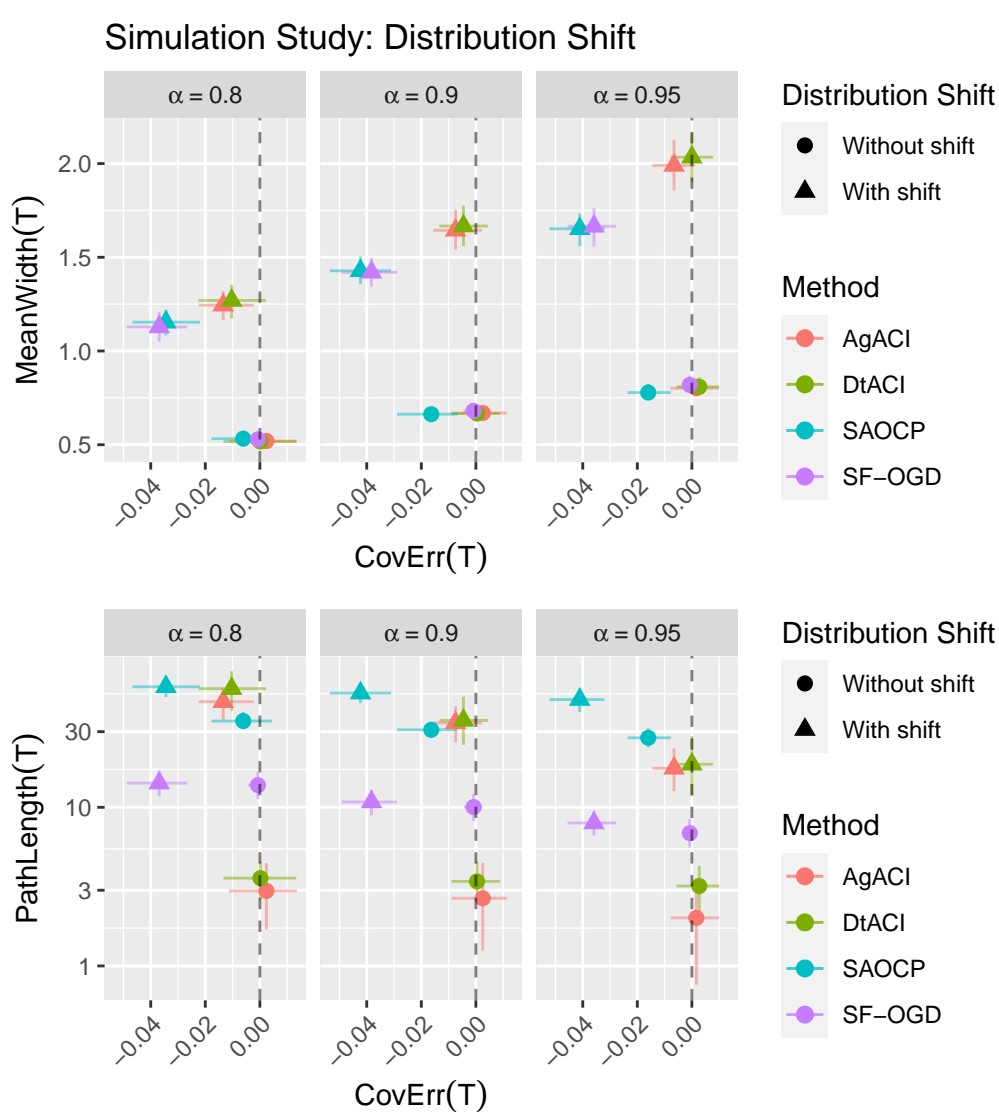

Figure 15: Mean interval width vs coverage error (top) and Mean Path Length vs. coverage error (bottom) for the second simulation study. The error bars represent the 10% to 90% quantiles of the metrics over the simulation datasets.

