# OpenReview forum: "AdaptiveConformal: An R Package for Adaptive Conformal Inference"
_Computo — Accepted by Computo_

### Review · Reviewer_kXB1 · 2024-04-09

**Summary Of Contributions:**

This paper introduces an R package dedicated to Adaptive Conformal Inference (ACI), that is derived from the conformal inference paradigm. Conformal inference gathers methods for uncertainty quantification that can be applied without any modeling assumption, except exchangeability of the data. ACI methods have been proposed in order to get through this last assumption and still be able to construct prediction intervals. As mentionned by the authors, ACI methods roughly relies on adjusting the size of next prediction sets depending on the previous ones covered or not the true observation, and the main theoretical difficulties remain in the tuning of some hyper-parameters.

After giving a quick theoretical framework in which the authors introduce in particular the considered type of prediction set and performance criteria, the authors give a nice overview for 4 ACI methods including their algorithm, theoretical guarantees of interest based on one (or most) of criteria introduced previously and a typical example to illustrate it. Then some simulation studies are proposed to compare the different methods under different frameworks before applying the methods to a real data set (influenza data in US from 2010 to 2017).

It is worth to notice though that even if the authors affirm that their main contribution is their package, I think that this work as it is proposed does not really reflect that claim.

**Audience:**

Yes

**Broader Impact Concerns:**

Not relevant.

**Claims And Evidence:**

No

**Requested Changes:**

## **General changes**
### *Major*
Concerning the code issues
- the functions run simulation study (1 and 2) are defined in the helpers.R file, but this is not precised anywhere.
- I needed to import by myself some dependencies (dplyr, tidyr and readr and ggplot2 to be more precise).
- The last command p19 returns `Error in plot coverr/plot piwidth : non-numeric argument to binary operator`
- Section 5 : the url linking to the data is not complete.

Please consider add some examples of usage of the package itself, for example by
- dedicating a (sub)section to it, based on the examples that are already available in the package
- providing in annex the codes producing the figures 1-7

### *Minor*
1. One possible issue when using ACI methods is to obtain trivial or infinite intervals (at least for ACI, AgACI and FACI), and it does not seem to be mentioned anywhere in the paper. The adopted solution(s), in particular for the implementation and the numerical illustration should appear.
2. For sake of readability, please consider postponing all the codes in annex.
3. Please precise the corresponding theorem when giving a theoretical guarantee. For example, coverage guarantee of sec 3.1.1 correspond to Proposition 4.1 in Gibbs and Candès (2021) (specific comment below)
4. Completing the Table 1 with theoretical guarantees for each algorithms would be welcome (or in another table)
5. As mentioned in the strengths, showing the typical behavior of the algorithms on a single trajectory is a good illustration. That could be nice to also get in annex an example of a pathological case (in which we would observe large undercoverage for example).

## **Specific comments**
1. p4, l5: $r_t=\\inf \\{ \\theta \\in \\mathbb{R} : y_t \\in \\hat{C}_t( \\theta) \\}$ (without the indicator function)
1. p4, l6-8: it could be worth to precize that the $D$ value is typically such that $\\hat{C}_t(D)$ correspond to the whole output space.
1. p4, l6-8: I guess the authors mean there exists a *finite* $D>0$. By the way, if $\\theta \\in[0;1]$ (i.e. for quantile intervals), does that means that $D$ is assumed to be $<1$?
1. Sec 2.2: It does not seem correct to define the general quantile interval function like that. It would be correct if the score function $S$ corresponds to a ``nice'' transformation of the residuals. That would be more general to consider the usual definition, that would be $\\hat{C}_t(\\theta_t) = \\{y : S(\\hat{\\mu}_t,y) \\leq Quantile(...)\\}$. It is possible then to get intervals that are not centered on $\hat{\mu}_t$ (or that don't even include it!).
1. Sec 3.1.1: the coverage guarantee is not exposed as in Gibbs and Candès (2021) ($\\max(\\theta_1,1-\\theta_1)$ instead of $D$) but as in Bhatnagar et al. (2023). Unless $D<1$, it seems a little less interesting written like that.
1. Sec 3.1.2: could the authors develop what is the ``doubling trick''? In same way as they summarize BOA in the next section. Or at least precise the section (2.3) of the cited book.
1. Sec 3.2.2: I would have say that AgACI tends to assign weight to lower values of $\\gamma$, which would make sens according to Fig 1
1. Fig 2 : the bottom-right misses two values of $\\gamma$ (0.064 and 0.128). Even if I guess it is because the points are close to 0, these points should appear. Also, the title indicates ``where the second grid is double the size of the first''.
1. Sec 3.3.2: the authors say that the $\\eta$ values is optimal, could you precise in which sense? CovErr or SAReg?
1. Fig 3: is it possible to get Figure 3 with at least another value of $\eta$? It would also be interesting to have similar visualisation of the weights as in Figure 2.
1. Sec 3.5: please give an interpretation of the parameter $g$ and/or about why the *lifetime* is defined like that, instead of another function. This can be found in Bhatnagar et al. (2023), but that would be welcome if it is described right here, especially as it does not require long explanation or reasoning.
1. It could be correct to comment that according to Fig 4 and Fig 5, both algorithms SAOCP and might behave poorly, compared to AgACi or FACI at least: the target coverage is achieved only when path length is really big (which seem confirmed by the simulation studis by the way).
1. Sec 4.1: 25 simulations seems way too low. I have been able to perform this simulation study in less than 5 seconds with $10^4$ simulations on my laptop (i5-10210U CPU 1.60GHz $\\times$ 8; 15.3 GiB RAM) without any issue.
1. Fig 6: path length graphs are hardly readable, because of SAOCP results. I would *suggest* to display it here without SAOCP, and postpone in annex the graph including SAOCP, or at least summarize
1. Sec 4.2: same remark as above: you could perform more than only 50 simulations for each type of time series. As an example, I performed this experiment with 500 simulations for each type in around 5 seconds on my laptop. It would requires way more computational time to consider $10^4$ simulations, but it seems reasonable to consider between 1000 and 5000 simulations.
1. Fig 8 is a nice visualization, but I would suggest to display here Figure 14 and put Figure 8 as an alternative plot for two reasons: 1. for sake of consistency with the previous simulation study and the further case study; 2. I find Fig 14 easier to read than Fig 8.
1. p26 l16: presentation instead of *presentaiton*
1. p26: the authors should precise which kind of predictors have been used *before* exposing Fig 10.
1. Sec 6: I would suggest to the authors to add a paragraph that would give a pro/cons for the 4 algorithms. The tuning parameters interpretability and their setting are well discussed, but this is actually illustrated with the examples based on a single trajectory only. Whereas the simulation experiments have been performed to study specific criteria (coverage error, intervals width and math length), which are not really discussed at the end.

**Strengths And Weaknesses:**

### **Strengths**
- As a non-english speaker, I am not able to judge the quality of the paper in terms of grammar or
syntax. However the paper remains nicely written and as well nice to read;
- Overall, the authors achieve to make their review accessible, even I think for a reader who would
not have any background on ACI;
- Overall, the visualizations are well chosen, typically for identifying the pro and cons for each
algorithm. It is in particular a good point to show the typical behavior of the algorithms on a
single trajectory.
### **Weaknesses**
- Reproducibility is not fully achieved, in particular because some codes exposed in the paper seem
to require modifications (see next section).
- Does not give some examples of the usage of the packages itself, which is quite a shame since it is
the title of the paper and is announced as the “primal practical contribution”.

---

> ### Author Response · Authors · 2024-06-13
> **Point-by-point responses (same order as in report)**
>
> ## Weaknesses
> - We have made updates to address this, detailed in the next section.
> - We have added a new section detailing usage of the package itself and pointing out the available documentation.
>
> ## Major points
> - We have added text to the README file explaining the role of helpers.R, and we have added comments to the code in the manuscript to indicate the definitions of run_simulation_study1 and run_simulation_study2 are to be found in helpers.R. We have added extra import statements to helpers.R so that it can be loaded and run independently of paper.qmd. We have updated how the URL is specified so that it is not cut off for being too long in the PDF.
> - We have added a section that describes the basic usage of the package with an example, and with information about where to find further documentation. All code used to generate the figures is available in either the paper.qmd or helpers.R files, which are both publicly available on Github.
>
> ## Minor points
>
> - We have added the following when introducing the quantile interval constructor: "Note we define $\widehat{C}_t(1) = \max\{s_1, \dots, s_{t-1}\}$ rather than $\widehat{C}_t(1) = \infty$ in order to avoid practical problems with trivial prediction intervals (Zaffran et al. 2022)."
> - We prefer to keep code in the main text for didactic reasons, but we note that in the HTML version, the code will be hideable which aids readability.
> - We have updated the manuscript to reference the corresponding theorems where appropriate.
> - We reformatted the table to accomodate more content, and added information about the theoretical guarantees for each algorithm.
> - We updated the single trajectory illustrations to use data generating process featuring a distribution shift, which does a better job showing some of the pathological situations that can arise (such as undercoverage, for example in Figure 1)
>
> ## Specific comments
> - So updated.
> - We have added a sentence to this effect: If the outcome space is bounded, then $D$ can be easily chosen to cover the entire space.
> - We have added the following note in the quantile section: "Note that we can always choose $D = 1$ to satisfy the outcome boundedness assumption given above.".
> - We have added this definition as an alternative with a comment on its interpretation.
> - We have updated Section 3.1.1 to express the coverage error bound both ways.
> - We have updated the citation to include the section of the cited book.
> - The results in Sec 3.2.2 have changed with the new simulation setup in the running example, and now the AgACI doesn't put weight on the lowest values of $\gamma$ in the second grid.
> - The caption has been corrected.
> - The value is optimal in terms of the dynamic regret, which we now clarify in the paper.
> - We have updated Figure 3 to show several choices of $\eta$, and commented on the results in the main text.
> - We have added a brief explanation and citation to explain the form of the lifetime function: "The form of the lifetime interval function L(t) is due to the use of geometric covering intervals to partition the input time series, and other choices may be possible (Jun et al. 2017)."
> - We added comments on this for both Figures 4 and 5.
> - We have updated the first simulation study to have 100 simulations, as increasing to e.g. 10\^4 simulations was computationally prohibitive on our computer (Macbook Air M1). Is it possible that when you ran 10\^4 in 5 seconds the code was loading the cached results of a previous simulation run?
> - We updated the path length plots in Figure 6 so that the y-axis is on the log10 scale in order to improve readability.
> - We have modified the second simulation study to have 100 simulations, matching the updated first simulation study.
> - We have updated the manuscript accordingly.
> - Fixed.
> - We prefer to treat how the individual forecasting methods worked, including what predictors they used (if any), as out of scope for this paper: "For our purposes, we treat the way the forecasts were produced as a black box."
> - We have added a paragraph to the discussion section.

---

> > ### Comment · Reviewer_kXB1 · 2024-07-08
> > **Response to revision**
> >
> > My apologies to the authors for this very delayed answer, due to a medical leave that ends today. I promise to get back to you within the next 2 days.

---

> > ### Comment · Reviewer_kXB1 · 2024-07-09
> >
> > I am fully satisfied by the answers and revisions made by the authors, and have recommended the paper to be accepted.

---

### Review · Reviewer_xpte · 2024-04-19

**Summary Of Contributions:**

This paper details a new R package containing implementations of a variety of adaptive conformal inference (ACI) algorithms. In addition, the paper performs simulation studies comparing the various methods and discusses the different theoretical guarantees offered by each approach. The paper is well written throughout and I believe that overall it offers a potentially valuable contribution to the community.

**Audience:**

Yes

**Broader Impact Concerns:**

I do not have any concerns about the ethical implementations of this work.

**Claims And Evidence:**

No

**Requested Changes:**

**Major points for revision:**
- The simulations would be greatly improved by the inclusion of more examples containing different types of distribution shift (e.g. gradual versus sharp changes). In particular, I think the running example in the paper of $y_t \sim N(0,0.2^2)$ is not the most insightful. Much more information about the methods would be gained if this example was replaced or paired with a more dynamic data generating process.
- The simulations should consider a notion of local or time-instantaneous performance of the methods. In particular, it would be helpful to know if the theoretical guarantees on the strongly adaptive regret given in previous works actually translate to improved local performance in practice.
- The discussion section makes a number of claims that I do not believe are clearly supported by the simulations in the article. These claims should either be weakened or supported by additional evidence. Specifically, 1) the authors suggest that ACI algorithms are sensitive to hyperparameter choices. However, for many of the algorithms in the paper no examples are shown of where the given hyperparameter choices fail., 2) The authors claim that "if AgACI does not perform well, one can simply increase the number of candidate learning rates." How would the user go about doing this? Would it always fix the performance?, 3) What is the difference between the simulations presented here and those in Bhatnagar et al. (2023)? Is there a reason the authors believe the present results are different from what was observed there?, 4) The authors refer to the sensitivity of ACI algorithms on the quality of the underlying point predictions. However, I don’t believe any of their simulations demonstrate this. Moreover, I’m not sure how aggregating machine learning models could fix this problem. Could the authors provide more information? 5) In the last sentence of the paper the authors reference assumptions that the outcome is bounded. However, many of the algorithms in this paper do not require bounded outcomes. Can the authors clarify?



**Minor points for revision:**
- The FACI algorithm has been renamed to DtACI.
- I believe the authors mean to refer to $m$, not $k$ in the definition of strongly adaptive regret on page 5.
- The definition of the pinball losses's subgradients on page 7 does not appear to match the definition of the loss on page 4.
- On page 7 the authors claim that an asymptotic coverage error of 0 can be obtained for any value of $\gamma$. However, this is not true if $\gamma$ is allowed to depend on $T$. This claim should be revised to reflect this.
- On the bottom of page 7, the authors should be careful when referring to $\gamma = D/\sqrt{T}$ as the “optimal choice”. This is only the optimal value with respect to long-term regret against constant alternatives. Against more sophisticated alternatives (dynamic regret, adaptive regret)  other choices can be optimal
- The authors appear to suggest that AgACI is unique in that it allows for two-sided estimates of the errors. However, I believe that all of the algorithms considered in this article can be readily adapted to estimate upper and lower bounds seperately.
- What is gamma at the beginning of section 3.3.1?
- There seems to be a typo in the coverage error bound at the top of page 12. Most notably, the exponential term looks like it should be inside the sum.
- It might be helpful to the reader to note that the final regret bound given for FACI on the middle of page 12 is a dynamic regret bound. This bound can easily be converted into a strongly adaptive regret bound by setting $\theta^*_t$ to a constant
- Can the authors give some insight into when one should expect the smoothness value $S_{\beta}(T)$ referenced on page 15 to be small?
- The authors use $w_t$ at various points to refer to both weights in the methods and widths of the intervals. Different letters would be preferable for clarity.
- It would be helpful to have some plots of what the data look like in Section 4.

**Strengths And Weaknesses:**

**Strengths:**
- The paper is well written throughout and all of the ACI methods are clearly explained.
- Simulation studies provide valuable insights into the utility of the various approaches.

**Weaknesses:**
- ACI algorithms are designed to handle distribution shift. Despite this, a significant fraction of the simulated datasets in the paper contain either no or minimal distribution shift.
- The authors extensively discuss the notion of strongly adaptive regret and its relation to the quality of the local performance of ACI algorithms. However, none of the simulations show the local performance of the methods.
- Many strong conclusions are made in the discussion section that are not fully supported by the results in the paper.
- The ACI literature is quite rich. While I understand that the authors cannot reasonably implement and give a detailed discussion of every method from the literature, I think the paper would benefit from a more comprehensive review that makes the reader aware of some of the other relevant works in this area.

---

> ### Author Response · Authors · 2024-06-13
> **Point-by-point responses (same order as in report)**
>
> ## Weaknesses:
> - Indeed the ACI was originally suggested in the context of distribution shifts, and since then ACI algorithms have been applied for more general settings where time series exhibit complex dependence structures. We have updated the running example used in the paper to exhibit a basic type of distribution shift as a way to better illustrate how the algorithms function.
> - We have updated the simulation studies to include a measure of the Strongly Adaptive Regret as a way to address the local performance of the algorithms.
> - We respond to the individual concerns about the discussion section in our response to the related comment under "Major points" below.
> - Indeed the ACI literature is rich and rapidly increasing, as several new preprints have appeared on the subject since we submitted the original manuscript. We have updated the introduction to include references to several relevant new works.
>
> ## Major points
> - We agree that a more dynamic data generating process would be informative, although there is nevertheless value in having a running example that is simple in order to introduce each method. We have therefore updated the running example so that the generated time series exhibits a sharp distribution shift: $y_t \sim N(0, \sigma_t^2)$ where $\sigma_t = 0.2$ if $t \leq 250$ and $\sigma_t = 0.1$ if $t > 250$.
> - We have updated the simulation studies to include the Strongly Adaptive Regret (with interval size m=20). In the second simulation study we found that, in the presence of distribution shift, SF-OGD and SAOCP tended to have smaller strongly adaptive regret than AgACI and DtACI. We have commented on this in the simulation study results.
> - We respond to each of the points separately:\
>     1) We have updated the claim in the discussion with the following to give a concrete example where a tuning parameter choice leads to underperformance: "for one example, Figure 4 shows how choosing the tuning parameter for SF-OGD to be too small can lead to intervals that update too slowly and significantly undercover."\
>     2) We have updated the comment about the robustness of AgACI to larger learning rate grids to reference the earlier example: "We also found that an advantage of the AgACI method is its robustness to the choice of its main tuning parameter, the set of candidate learning rates, in the sense that the grid of candidate learning rates can always be expanded as illustrated in 3.2.2".\
>     3) The time-series forecasting experiments presented in Bhatnagar et al. (2023) are valuable but differ from the simulations in our paper in that they are based on real-world observed time-series (from the M4 Competition and the NN5 dataset), while our results are based on simulated time-series following known data-generating processes.\
>     4) We have updated this paragraph to explain in more detail how aggregation methods may improve the quality of point predictions used as input to the ACI algorithms.
>     5) We have updated the final sentence to read "Another avenue for theoretical research is to relax the assumption of bounded radii necessary for the theoretical results of algorithms such as SAOCP."
>
> ## Minor points
>
> - We have updated the paper to use the new name DtACI.
> - Yes, we have corrected this.
> - The definition of the pinball loss has been corrected.
> - We have added a note to this effect.
> - We have updated the text to specify in what respect this choice is optimal.
> - We agree that the algorithms could be extended simply to estimate the upper and lower bounds separately; this is currently pointed out in the discussion in this sentence: "A simple extension would switch to using the interval loss function (Gneiting and Raftery 2007), which would allow for asymmetric intervals where two parameters are learned for the upper and lower bounds, respectively"
> - Gamma in this equation was a typo, and should instead be eta, a meta learning parameter of the DtACI algorithm. The text has been updated to correct this.
> - The coverage error bound formula has been updated.
> - We have added a sentence noting this: "This dynamic regret bound can be converted to a strongly adaptive regret bound by choosing $\theta^*_t$ to be constant."
> - We have added an additional sentence to explain when the smoothness value may be small.
> - We have updated the algorithm descriptions to use $p_t$ rather than $w_t$ to refer to expert weights.
> - We have added an additional figure that shows a subset of the data for one realization of the simulation data generating process.

---

> > ### Comment · Reviewer_xpte · 2024-06-25
> > **Response to revision**
> >
> > I appreciate the work done by the authors to revise the manuscript. All of my concerns have been adequately addressed and I have recommended that the paper be accepted.

---

### Comment · Action_Editor_8kr6 · 2024-05-06
**Delay for revision**

Dear reviewers,
Thank you for reviewing this papers. The authors need a few more weeks to submit a revised version addressing your concerns. You may have received an email asking you to submit your official recommendation : please wait until the authors have submitted their revision to do so.
Best regards,
AC

---

### Comment · Action_Editor_8kr6 · 2024-07-18
**Lifting anonymity**

Dear reviewers,
Thank you again for reviewing this paper for Computo.
We are currently in the process of publishing it. If you want to get recognition for it, we can deanonymize your review.
Can you answer here to let me know if you want your identity to be public?
Best,
Mathurin

---

> ### Comment · Reviewer_xpte · 2024-07-23
> **Staying anonymous**
>
> I'd prefer my review stay anonymous.

---

> ### Comment · Reviewer_kXB1 · 2024-07-24
>
> I would also prefer to stay anonymous! Best.

---

### Comment · Action_Editor_8kr6 · 2024-07-18
**Proofreading before publication**

Dear authors,
Your paper is ready for publication at https://computo.sfds.asso.fr/published-202407-susmann-adaptive-conformal/published-202407-susmann-adaptive-conformal.pdf
Should you notice any typo (in particular, in affiliations etc), can you send a PR to https://github.com/computorg/published-202407-susmann-adaptive-conformal ?
Best,
The computo editorial board

---

### Note · Reviewer_xpte · 2024-06-25

**Comment:**

This paper details a new R package containing implementations of a variety of adaptive conformal inference (ACI) algorithms and compares their performance on a variety of datasets. I believe that this work will be of interest to the readership of computo and that the authors have done well to address the concerns raised by myself and the other reviewer.

**Audience:**

Yes

**Claims And Evidence:**

Yes

**Decision Recommendation:**

Accept

---

### Note · Reviewer_kXB1 · 2024-07-09

**Comment:**

I am fully satisfied by the answers and revisions made by the authors.

**Audience:**

Yes

**Claims And Evidence:**

Yes

**Decision Recommendation:**

Accept

---

### Decision · Action_Editor_8kr6 · 2024-07-09

**Recommendation:** Accept as is

**Comment:**

n/a

**Audience:**

yes

**Claims And Evidence:**

yes

---

> ### Decision · Editors_In_Chief · 2024-07-09
>
> I approve the AE's decision.